# FedEx-LoRA: Exact Aggregation for Federated and Efficient Fine-Tuning of Foundation Models

## Abstract

Low-Rank Adaptation (LoRA) is a popular technique for efficient fine-tuning of foundation models. However, applying LoRA in federated learning environments, where data is distributed across multiple clients, presents unique challenges. Existing methods rely on traditional federated averaging of LoRA adapters, resulting in inexact updates. To address this, we propose **Fed**erated **Ex**act **LoRA**, or **FedEx-LoRA**, which adds a residual error term to the pretrained frozen weight matrix. Our approach achieves exact updates with minimal computational and communication overhead, preserving LoRA's efficiency. We evaluate the method on various models across arithmetic reasoning, commonsense reasoning, natural language understanding and natural language generation tasks, showing consistent performance gains over state-of-the-art methods across multiple settings. Through extensive analysis, we quantify that the deviations in updates from the ideal solution are significant, highlighting the need for exact aggregation. Our method's simplicity, efficiency, and broad applicability position it as a promising solution for accurate and effective federated fine-tuning of foundation models.

## 1 Introduction

The introduction of large language models (LLMs) has revolutionized natural language processing, enabling unprecedented performance across a wide range of tasks (Achiam et al., 2023; Touvron et al., 2023; Team et al., 2023; Chang et al., 2024; Raffel et al., 2020; Zeng et al., 2022). While these models excel at transfer learning, their true potential is often unlocked through fine-tuning — a critical process that aligns these general-purpose models with specific tasks or domains. Moreover, the sheer size of these models presents significant challenges for fine-tuning and deployment, particularly in resource-constrained or distributed environments. To address these challenges, parameter-efficient fine-tuning (PEFT) methods have gained prominence, with Low-Rank Adaptation (LoRA) emerging as a particularly effective approach (Hu et al., 2021). LoRA's success lies in its ability to adapt LLMs to new tasks by training only a small number of parameters, while freezing rest of the parameters. This significantly reduces computational and memory requirements without compromising performance. Although good progress in training of LLMs has been realized by entities equipped with massive computational resources, there is hoards of unreachable data in verticals such as healthcare, finance, law firms, social-media and logistics. Federated learning (FL) is a popular paradigm to learn a machine learning model in this setting with multiple distributed entities (Konečný et al., 2017; Kairouz et al., 2021; Bonawitz et al., 2019) holding siloed data.

Federated Fine-Tuning (FFT) for foundation models addresses the challenge of leveraging distributed datasets while preserving data privacy. The current state-of-the-art, Federated Instruction Tuning (FedIT, Zhang et al. (2024b)), uses conventional federated aggregation to average the low-rank matrices $\mathbf{A}$ and $\mathbf{B}$ individually. The resulting update matrix which is formed post aggregation is thus the product of the averaged matrices $\mathbf{A}$ and $\mathbf{B}$. However, the ideal update should be the average of the products of the low-rank adapters $\mathbf{A}$ and $\mathbf{B}$. The discrepancy results from the fact that *"the average of the products is not equal to the product of the averages"*. A naive adhoc intervention of modifying the aggregation to directly average the client updates is not a viable solution, since the subsequently obtained weight matrix loses its low-rank structure. The low-rank structure provides the efficiency benefits of LoRA in the first place, making this approach computationally intractable.

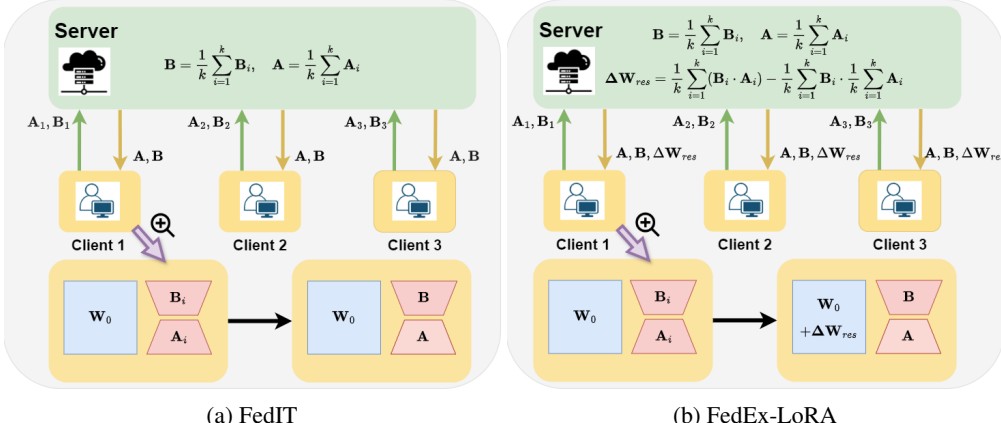

|     |     |
| :-: | :-: |
| (a) FedIT | (b) FedEx-LoRA |

Figure 1: Comparison of federated LoRA methods: (a) **FedIT** averages the individual client low-rank adapters $\mathbf{A}_i$ and $\mathbf{B}_i$, resulting in inexact updates. (b) **FedEx-LoRA** sends the error residual $\Delta\mathbf{W}_{res}$ along with the individual adapters $\mathbf{A}_i$ and $\mathbf{B}_i$, which is added to the pretrained weight matrix $\mathbf{W}_0$, ensuring exact aggregation. Clients transmit low-rank adapters $\mathbf{A}_i$ and $\mathbf{B}_i$ in both methods.

The aggregation process must be carefully designed for both accuracy and simplicity. We introduce FedEx-LoRA, a method that improves federated aggregation for LoRA by incorporating an error residual term, $\Delta\mathbf{W}_{res}$, into the pretrained weight matrix to address inexact aggregation, as shown in Figure 1. This adjustment preserves the low-rank efficiency of LoRA without adding computational overhead. Since the average update is inherently higher rank and cannot fit into the low-rank adapters, it is absorbed into the pretrained weight matrix, which is already high rank. This error term requires no training and is added at each aggregation step, ensuring no additional training costs.

Our key contributions are summarized as follows:

- We identify a critical discrepancy in traditional federated averaging of LoRA adapters and address it by explicitly assigning the error residual to the pretrained weight matrix, ensuring ideal updates.

- The error residual term is incorporated at each aggregation step, maintaining LoRA's efficiency without any additional training. We propose a communication protocol that minimizes both communication and computational overhead.

- We demonstrate the effectiveness of our approach through extensive experiments on models ranging from RoBERTa-base (125M) to Gemma-2 (9B) across arithmetic reasoning, commonsense reasoning, natural language understanding, and generation tasks. Our method consistently outperforms state-of-the-art federated fine-tuning techniques, showing clear performance gains.

- We provide a detailed analysis of the deviations introduced by federated averaging compared to ideal updates, and identify notable patterns. We further show that while multiple assignment strategies exist for exact aggregation, our specific assignment approach is most effective.

## 2  RELATED WORK

**Parameter-efficient Fine-tuning.** PEFT methods aim to adapt foundation models while minimizing the number of trainable parameters. Input-based techniques like prefix tuning (Li & Liang, 2021) prepend trainable prompts, and prompt tuning (Lester et al., 2021) optimizes soft prompts in the embedding space - both effective for task-specific adaptations. Architectural approaches, such as adapter layers (Houlsby et al., 2019), add trainable components between transformer blocks (Vaswani et al., 2017), facilitating multi-task learning. LoRA (Hu et al., 2021) reduces memory overhead by representing weight updates with low-rank matrices, while AdaLoRA (Zhang et al., 2023b) improves efficiency by dynamically adjusting the parameter budget. Optimization techniques, like QLoRA (Dettmers et al., 2024), enable fine-tuning on consumer hardware via quantization, and LongLoRA (Chen et al., 2024) targets long-context tasks. Recent advancements include combining multiple

PEFT methods (Lin et al., 2024) and scaling these techniques for very large models (Zhang et al., 2024a), advancing the state of efficient fine-tuning.

**Federated Fine-Tuning of Foundation Models.** Federated learning (Konečný et al., 2017) is a decentralized approach that allows multiple clients to collaboratively train a shared model without sharing their private data. Instead, clients perform local training on their own datasets, and only the resulting model updates are securely aggregated to update the global model (Kairouz et al., 2021). This iterative process of local training and global aggregation continues until the model converges. FedBERT (Tian et al., 2022) introduced federated pre-training for BERT, while recent efforts have focused on federated fine-tuning of foundation models (Zhang et al., 2022; Kuang et al., 2024; Babakniya et al., 2023). The current state-of-the-art, FedIT (Zhang et al., 2024b)), fine-tunes LLMs by averaging LoRA parameters across clients using vanilla Federated Averaging (FedAvg, McMahan et al. (2017)). However, averaging low-rank adapters independently introduces noise and results in inexact global updates. Federated Freeze A LoRA (FFA-LoRA) (Sun et al., 2024) mitigates this by keeping one set of adapters trainable, improving aggregation stability but limiting the training flexibility of other adapters. This method is particularly advantageous in privacy-sensitive settings (Huang et al., 2022; Zhang et al., 2021). Another challenge arises from heterogeneous rank settings, where clients adjust LoRA ranks based on their capacities (Zhao et al., 2018; Li et al., 2019). Some methods address this by self-pruning local LoRA modules and employing sparsity-weighted aggregation (Cho et al., 2024), though this introduces substantial computational overhead.

## 3 PRELIMINARIES AND MOTIVATION

**Fine-tuning with LoRA.** LoRA (Hu et al., 2021) leverages low-rank matrix factorization to efficiently represent the updates of pre-trained model weights. Specifically, the fine-tuned weights, $\mathbf{W}'$, are expressed as a sum of the original weights $\mathbf{W_0}$ and a low-rank update $\mathbf{\Delta W}$:

$$\mathbf{W}' = \mathbf{W_0} + \mathbf{\Delta W} = \mathbf{W_0} + \mathbf{B}\mathbf{A} \tag{1}$$

where $\mathbf{W_0}, \mathbf{W}' \in \mathbb{R}^{m \times n}$ are the pretrained and fine-tuned weight matrices, respectively, and $\mathbf{A} \in \mathbb{R}^{r \times n}$, $\mathbf{B} \in \mathbb{R}^{m \times r}$ represent the low-rank decomposition of $\mathbf{\Delta W}$. Here, the rank $r$ is significantly smaller than both $m$ and $n$, leading to a substantial reduction in the number of trainable parameters for $\mathbf{\Delta W}$. Instead of directly updating $\mathbf{W_0}$ during fine-tuning, LoRA optimizes the smaller matrices $\mathbf{A}$ and $\mathbf{B}$, resulting in considerable savings in memory usage. For instance, in GPT-2, LoRA reduces the number of trainable parameters from $124.44$ M to just $0.41$ M when using a rank of $r = 4$, with no observed degradation in performance (Hu et al., 2021).

**Global Updates due to Vanilla Federated Averaging are Inexact.** The widely adopted federated learning algorithm, FedAvg (McMahan et al., 2017), updates the global model by performing a weighted average of local client updates in each communication round for $k$ clients:

$$\mathbf{W}^{global} = \mathbf{W_0} + \frac{1}{k}\sum_{i=1}^{k}\mathbf{\Delta W}_i = \mathbf{W_0} + \mathbf{\Delta W} \tag{2}$$

where $\mathbf{W_0}$ and $\mathbf{W}^{global}$ represent the global model parameters before and after aggregation, respectively. $\mathbf{\Delta W}_i$ denotes the local update from the $i$-th client. FedIT (Zhang et al., 2024b) extends FedAvg by incorporating LoRA for federated fine-tuning, where clients fine-tune LoRA modules of a fixed rank. The global LoRA matrices $\mathbf{A}$ and $\mathbf{B}$ are updated via weighted averaging over the client-specific LoRA parameters $\mathbf{A}_k$ and $\mathbf{B}_k$:

$$\mathbf{A} = \frac{1}{k}\sum_{i=1}^{k}\mathbf{A}_i, \quad \mathbf{B} = \frac{1}{k}\sum_{i=1}^{k}\mathbf{B}_i \tag{3}$$

Although FedIT follows a similar aggregation process as FedAvg, only LoRA modules are updated and communicated. However, this independent averaging of $\mathbf{A}_i$ and $\mathbf{B}_i$ introduces deviation from the exact centralized LoRA updates, as the actual model updates depend on the product $\mathbf{B}_i\mathbf{A}_i$, not the individual components $\mathbf{B}$ and $\mathbf{A}$.

$$\underbrace{\tilde{\mathbf{W}}^{global} = \mathbf{W_0} + \frac{1}{k}\sum_{i=1}^{k}\mathbf{B}_i \times \frac{1}{k}\sum_{i=1}^{k}\mathbf{A}_i}_{\text{Parameters after aggregation with LoRA + FedAvg (FedIT)}} \neq \underbrace{\mathbf{W_0} + \frac{1}{k}\sum_{i=1}^{k}(\mathbf{B}_i\mathbf{A}_i) = \mathbf{W}^{global}}_{\text{Ideal parameters following model-averaging}} \tag{4}$$

**There is No Free Lunch.** A naive approach would be to directly average the client updates as $\frac{1}{k}\sum_{i=1}^{k}(\mathbf{B}_i\mathbf{A}_i)$ and use the result for the global update before resuming training. However, this undermines the purpose of LoRA, as it forces subsequent training on the full-rank matrix $\mathbf{W}^{global} \in \mathbb{R}^{m \times n}$ rather than its intended low-rank adapters $\mathbf{A} \in \mathbb{R}^{r \times n}$ and $\mathbf{B} \in \mathbb{R}^{m \times r}$.

An alternative is to decompose the averaged update $\frac{1}{k}\sum_{i=1}^{k}(\mathbf{B}_i\mathbf{A}_i)$ into a low-rank matrix of rank $(k \cdot r)$. However, this leads to an exponential growth in the rank with each aggregation round, as the rank increases by a factor of $k$ in every iteration, making this approach computationally intractable.

**FFA-LoRA.** FFA-LoRA addresses the problem of inexact aggregation, particularly in privacy-preserving settings. Motivated from previous works (Zhang et al., 2023a; Tian et al., 2024), it asymmetrically freezes the $\mathbf{A}$ adapters while keeping only the $\mathbf{B}$ adapters trainable. This approach mitigates the issues of non-ideal aggregation by avoiding independent updates of $\mathbf{A}$ and $\mathbf{B}$. However, the drawback is that the $\mathbf{A}$ matrix remains static, which limits expressiveness. While this method excels in privacy-sensitive scenarios where noise is amplified, it underperforms in non-private settings, even when the number of trainable parameters is equivalent.

## 4 METHOD: FEDEX-LORA

### 4.1 NOISE-FREE EXACT AGGREGATION

To tackle the problem of inexact aggregation arising from the independent averaging of the $\mathbf{A}$ and $\mathbf{B}$ matrices across clients, we introduce a novel method called FedEx-LoRA. Instead of separately averaging the low-rank adapter matrices $\mathbf{A}$ and $\mathbf{B}$, we compute the average of their product $\mathbf{BA}$ across all clients. However, as previously noted in Section 3, we cannot keep this high-rank matrix or its lower-rank decomposition (with rank $(k \cdot r)$) trainable. Consequently, we append a high-rank error term that captures the discrepancy between the average of the products and the product of the averages. This error residual is incorporated into the global frozen weight matrix, ensuring its non-trainability. The update at the $j^{th}$ aggregation round can be expressed as follows:

$$\mathbf{B}_i^{j+1} \leftarrow \frac{1}{k}\sum_{i=1}^{k}\mathbf{B}_i^j, \quad \mathbf{A}_i^{j+1} \leftarrow \frac{1}{k}\sum_{i=1}^{k}\mathbf{A}_i^j \tag{5}$$

$$\mathbf{W_0}^{j+1} \leftarrow \mathbf{W_0}^j + \underbrace{\frac{1}{k}\sum_{i=1}^{k}(\mathbf{B}_i^j\mathbf{A}_i^j) - \frac{1}{k}\sum_{i=1}^{k}\mathbf{B}_i^j \times \frac{1}{k}\sum_{i=1}^{k}\mathbf{A}_i^j}_{\text{Residual}} \tag{6}$$

We now demonstrate that our formulation results in exact aggregation for every client:

$$\mathbf{W}_{global}^{j+1} = \mathbf{W_0}^j + \mathbf{B}_i^j\mathbf{A}_i^j \tag{7}$$

$$\mathbf{W}_{global}^{j+1} = \mathbf{W_0}^j + \frac{1}{k}\sum_{i=1}^{k}(\mathbf{B}_i^j\mathbf{A}_i^j) - \frac{1}{k}\sum_{i=1}^{k}\mathbf{B}_i^j \times \frac{1}{k}\sum_{i=1}^{k}\mathbf{A}_i^j + \frac{1}{k}\sum_{i=1}^{k}\mathbf{B}_i^j \times \frac{1}{k}\sum_{i=1}^{k}\mathbf{A}_i^j \tag{8}$$

$$\mathbf{W}_{global}^{j+1} = \mathbf{W_0}^j + \underbrace{\frac{1}{k}\sum_{i=1}^{k}(\mathbf{B}_i^j\mathbf{A}_i^j)}_{\text{Ideal aggregation}} \tag{9}$$

### 4.2 FEDEX-LORA: OVERALL PIPELINE

Initially, the server distributes the global pretrained model to all $k$ clients and initializes the low-rank adapters $\mathbf{A}$ and $\mathbf{B}$ according to standard LoRA settings: $\mathbf{B}$ is initialized to zero, while $\mathbf{A}$ is initialized using a random Gaussian distribution.

$$\mathbf{B}_i^0 \leftarrow \mathbf{B}_{init}, \quad \mathbf{A}_i^0 \leftarrow \mathbf{A}_{init}, \quad \mathbf{W}_0^0 \leftarrow \mathbf{W}_{pretrained} \tag{10}$$

Each client then independently trains their low-rank adapters $\mathbf{A}$ and $\mathbf{B}$ using their local data for a specified number of epochs (referred to as "local epochs"). Upon completion of training, the clients

send their updated low-rank adapters back to the server for aggregation. The server aggregates these low-rank adapters and incorporates the residual term into the global model:

$$\mathbf{B}_{global}^j = \frac{1}{k}\sum_{i=1}^{k}\mathbf{B}_i^j, \quad \mathbf{A}_{global}^j = \frac{1}{k}\sum_{i=1}^{k}\mathbf{A}_i^j \tag{11}$$

$$\boldsymbol{\Delta}\mathbf{W}_{res}^j = \frac{1}{k}\sum_{i=1}^{k}(\mathbf{B}_i^j\mathbf{A}_i^j) - \frac{1}{k}\sum_{i=1}^{k}\mathbf{B}_i^j \times \frac{1}{k}\sum_{i=1}^{k}\mathbf{A}_i^j \tag{12}$$

The server then sends the aggregated matrices back to each client. After receiving these updates, the clients proceed to update their low-rank adapters $\mathbf{A}$ and $\mathbf{B}$, as well as the weight matrix:

$$\mathbf{B}_i^{j+1} \leftarrow \mathbf{B}_{global}^j, \quad \mathbf{A}_i^{j+1} \leftarrow \mathbf{A}_{global}^j \tag{13}$$

$$\mathbf{W}_0^{j+1} \leftarrow \mathbf{W}_0^j + \boldsymbol{\Delta}\mathbf{W}_{res}^j \tag{14}$$

Following this, clients independently resume fine-tuning for a set number of local epochs. This process repeats across multiple aggregation rounds (also referred to as communication rounds).

**Multiple Assignment Strategies can Lead to Exact Aggregation.** Several methods can be used for achieving exact aggregation, with our choice of assignments for $\mathbf{A}_i$ and $\mathbf{B}_i$ being particularly pivotal. Each such assignment strategy allows us to adjust the corresponding error offset within the frozen weight matrix, facilitating precise aggregation. In Section 6, we investigate various methods and empirically show that our proposed assignments for $\mathbf{A}_i$ and $\mathbf{B}_i$ deliver the best performance.

**Communication Protocol.** At first glance, it may seem necessary for the server to transmit the high-rank update matrix $\boldsymbol{\Delta}\mathbf{W}_{res}$ to the clients, which could introduce substantial communication overhead. However, the rank of this update matrix is capped at $(k \cdot r)$. Consequently, $\boldsymbol{\Delta}\mathbf{W}_{res}$ can be decomposed into two low-rank matrices using methods such as Gram-Schmidt orthogonalization. This decomposition expresses the matrix as a product of the basis of its column (or row) space and the corresponding linear coefficients. The *computational* overhead incurred by this operation at each aggregation step is negligible compared to the numerous matrix multiplications involved in training. Importantly, clients are only required to transmit their low-rank adapters $\mathbf{A}_i$ and $\mathbf{B}_i$, avoiding the need to send any high-rank update matrices. In practice, the *communication* overhead is minimal compared to FedIT, and overall, the *communication* cost remains significantly lower than that of full federated fine-tuning. Detailed communication overhead analysis is provided in Section 6.

**Best Inexact Approximation.** For exact aggregation, the communication cost scales linearly with the number of clients, becoming prohibitive in hyperclient settings. To address this, we propose relaxing the exact aggregation condition through truncated SVD of the residual matrix. This reconstruction yields a low-rank approximation which, by the Eckart-Young theorem (Eckart & Young, 1936), is provably optimal for the high-rank update matrix. Specifically, for a target rank $r'$, the best low-rank approximation $\Delta W_{rec}^{r'}$ is computed as:

$$U, S, V^T \leftarrow \mathbf{SVD}(\Delta W_{res}) \tag{15}$$

$$\Delta W_{rec}^{r'} \leftarrow U[1:r']S[1:r', 1:r']V^T[1:r'] \tag{16}$$

While this method introduces approximation error, it provides the theoretically optimal approximation to exact aggregation. A key advantage is that the server can control communication costs, a capability absent in previous methods - FedIT (Zhang et al., 2024b) and FFA-LoRA (Sun et al., 2024).

## 5 EXPERIMENTS

**Models and Datasets.** We evaluate our method on four NLP benchmarks using models ranging from RoBERTa-base with 125M parameters to Gemma-2 with 9B parameters, covering both masked and autoregressive architectures. Our experiments include fine-tuning Mistral-7B (Jiang et al., 2023), Gemma-2 9B (Team et al., 2024), Llama-3.2 3B (Dubey et al., 2024), RoBERTa-base, RoBERTa-large (Liu et al., 2019), and GPT-2 (Radford et al., 2019) using FedEx-LoRA. This comprehensive setup allows us to assess the effectiveness of our approach across different tasks and model architectures.

For arithmetic reasoning, we fine-tune the decoder-only models Mistral-7B and Gemma-2 9B using 10K samples from the MetaMathQA dataset (Yu et al., 2024). These models are evaluated on two standard arithmetic reasoning benchmarks, GSM8K (Cobbe et al., 2021) and MATH (Hendrycks et al., 2021). In the commonsense reasoning category, we use Llama-3.2 3B, which is trained on COMMONSENSE170K—a compilation of eight commonsense reasoning datasets (Hu et al., 2023). We evaluate the RoBERTa models on natural language understanding tasks with the GLUE benchmark (Wang et al., 2019) and assess GPT-2 on natural language generation tasks through the E2E NLG Challenge (Novikova et al., 2017). We implement all algorithms using PyTorch (Paszke et al., 2019), based on the widely-used HuggingFace Transformers codebase (Wolf et al., 2020). We run all experiments on a single NVIDIA A100/A6000 GPU, and present the results as average of 3 different random runs. Base models are loaded in `torch.bfloat16` to save memory. Dataset details are presented in Appendix A.

**Implementation Details.** The residual and product matrices are scaled by the factor $\alpha/r$, where $\alpha$ is a constant in $r$, consistent with the approach in LoRA (Hu et al., 2021). We run our experiments in a three-client cross-silo federated setting, based on the settings described in FFA-LoRA (Sun et al., 2024). For data distribution among clients, we use the common method to sample data at random for each client, as implemented in standard works (Zhang et al., 2024b; He et al., 2020; Lai et al., 2022).

**Baselines.** We primarily compare FedEx-LoRA with other federated fine-tuning versions of LoRA, but include centralized LoRA as a *performance benchmark* or *skyline*. We also include other baselines, where possible. **Full Fine-Tuning (FT)** refers to fine-tuning the entire pretrained model. **LoRA** (Hu et al., 2021) represents the traditional centralized LoRA approach. **FedIT** (Zhang et al., 2024b), the current state-of-the-art federated fine-tuning method, applies vanilla federated averaging (FedAvg) to LoRA (McMahan et al., 2017). **FFA-LoRA** (Sun et al., 2024) freezes the **A** matrices and trains only the **B** matrices, allowing for exact aggregation in a federated setting but at the cost of losing the benefits of training **A**.

## 5.1 INSTRUCTION TUNING

**Implementation Details.** For **arithmetic reasoning**, we fine-tune Mistral-7B (Jiang et al., 2023) and Gemma-2 9B (Team et al., 2024) on 10K samples from the MetaMathQA dataset (Yu et al., 2024) and evaluate them on the GSM8K (Cobbe et al., 2021) and MATH (Hendrycks et al., 2021) benchmarks. For **commonsense reasoning**, we use Llama-3.2 3B, training it on COMMONSENSE170K—a dataset combining eight commonsense reasoning datasets (Hu et al., 2023)—and evaluate its performance on each of those datasets. In all instruction tuning tasks, we apply LoRA modules to the key, value, query, attention output, and all fully connected weight matrices. We fine-tune over a single local epoch within one aggregation round, using a rank of $r = 32$.

**Main Results.** Tables 1 and 2 present the results for commonsense and arithmetic reasoning. Our method consistently surpasses state-of-the-art federated fine-tuning techniques across both arithmetic benchmarks and all eight commonsense reasoning tasks for every evaluated model. For example, on average accuracy for commonsense reasoning, FedEX-LoRA outperforms FFA-LoRA by $8.63\%$ and FedIT by $2.42\%$ respectively.

| Method | Accuracy ($\uparrow$) | | | | | | | | |
|---|---|---|---|---|---|---|---|---|---|
| | BoolQ | PIQA | SIQA | HellaS. | WinoG. | ARC-e | ARC-c | OBQA | Avg. |
| Centralized LoRA$_{r=32}$ | 73.45 | 89.65 | 82.23 | 94.41 | 87.97 | 93.88 | 82.76 | 86.60 | 86.37 |
| FedIT$_{r=32}$ | 70.73 | 87.59 | 79.17 | 91.06 | 83.42 | 92.71 | 81.31 | 82.68 | 83.57 |
| FFA-LoRA$_{r=32}$ | 65.78 | 84.22 | 72.41 | 82.27 | 72.53 | 90.36 | 76.28 | 75.00 | 77.35 |
| FedEx-LoRA$_{r=32}$ | **73.21** | **89.01** | **81.98** | **94.29** | **87.29** | **93.68** | **82.33** | **86.20** | **85.99** |

Table 1: Results for Llama-3.2 3B on eight commonsense reasoning datasets, comparing various federated LoRA methods at rank $r = 32$. **Centralized LoRA (in grey) sets the benchmark skyline** for its federated versions. Best results among federated methods (in blue) are highlighted in **bold** for each setting.

| Model | Method | Accuracy (↑) | |
|---|---|---|---|
| | | **GSM8K** | **MATH** |
| Mistral-7B | Centralized LoRA$_{r=32}$ | 62.77 | 16.24 |
| | FedIT$_{r=32}$ | 56.94 | 14.96 |
| | FFA-LoRA$_{r=32}$ | 56.41 | 14.88 |
| | FedEx-LoRA$_{r=32}$ | **62.62** | **16.54** |
| Gemma-2 9B | Centralized LoRA$_{r=32}$ | 76.34 | 39.32 |
| | FedIT$_{r=32}$ | 74.57 | 37.16 |
| | FFA-LoRA$_{r=32}$ | 75.04 | 35.18 |
| | FedEx-LoRA$_{r=32}$ | **76.19** | **39.00** |

Table 2: Arithmetic reasoning performance on GSM8K and MATH for Mistral-7B and Gemma-2 9B, comparing various federated LoRA methods at rank $r = 32$. **Centralized LoRA (in grey) sets the benchmark skyline** for its federated versions. Best results among federated methods (in blue) are highlighted in **bold** for each setting.

## 5.2 NATURAL LANGUAGE UNDERSTANDING

**Implementation Details.** RoBERTa (Liu et al., 2019) is a widely used pretrained model known for its competitive performance among its size. We use the pretrained RoBERTa-base (125M parameters) and RoBERTa-large (355M parameters) from the HuggingFace Transformers library (Wolf et al., 2020) and evaluate them on several datasets from the GLUE benchmark: CoLA, RTE, MRPC, SST-2, QNLI, and STS-B. We apply LoRA modules only to the self-attention layers, following the setup from the original LoRA paper (Hu et al., 2021). Models are fine-tuned at ranks $r = \{4, 1\}$ over local epochs of 3 and 10. For RoBERTa-base, we run 50 aggregation rounds for 3 local epochs and 15 rounds for 10 local epochs. For RoBERTa-large, we perform 15 aggregation rounds for 3 local epochs and 5 rounds for 10 local epochs. Detailed experimental settings are provided in Appendix B.

**Main Results.** We present results for RoBERTa-base and RoBERTa-large in Table 3, evaluated at ranks $r = \{4, 1\}$. Our method consistently outperforms state-of-the-art federated fine-tuning approaches across all datasets and settings. Notably, our method occasionally achieves performance on par with centralized LoRA. Additional results in Appendix D (Table 10) further demonstrate the robustness and superiority of our method over other federated LoRA variants across multiple settings.

## 5.3 NATURAL LANGUAGE GENERATION

**Implementation Details.** We fine-tune GPT-2 (124M parameters) (Radford et al., 2019) on the E2E NLG Challenge dataset (Novikova et al., 2017). We apply LoRA modules only to the self-attention layers. The model is fine-tuned at ranks $r = \{4, 1\}$ with local epochs set to 3 and 10, using 6 aggregation rounds for both settings. Detailed experimental settings are provided in Appendix B.

**Main Results.** Table 4 presents the performance of GPT-2 fine-tuned with ranks $r = \{4, 1\}$. FedEx-LoRA consistently outperforms leading federated fine-tuning methods, across all metrics and settings. Additional evaluations, provided in Appendix E (Table 11), further demonstrate the reliability and strength of FedEx-LoRA across different configurations.

## 6 ANALYSIS

To fully understand the implications of our method, we performed several in-depth analyses, each targeting a specific aspect of FedEx-LoRA's performance and efficiency.

**Assignment Strategies for $\mathbf{A}_i$ and $\mathbf{B}_i$.** As discussed in Section 4, we can incorporate any high-rank update matrix $\mathbf{\Delta W}_{res}$ within the frozen full-rank matrix $\mathbf{W}_0$. However, assignment of the low-rank adapters $\mathbf{A}_i$ and $\mathbf{B}_i$ post-aggregation is less straightforward. Any selection of $\mathbf{A}_i$ and $\mathbf{B}_i$ can be offset by adjusting the residual update, by ensuring that $\mathbf{W}_0 + \mathbf{B}_i \mathbf{A}_i$ remains consistent across clients. We evaluate three strategies: (1) **Reinitialize $\mathbf{A}_i$ and $\mathbf{B}_i$** reinitializes $\mathbf{A}_i$ and $\mathbf{B}_i$ after aggregation and appends the full update to the frozen weights (ensuring $\mathbf{W}_0 + \mathbf{B}_i \mathbf{A}_i$ is identical). (2) $\mathbf{A}_i \leftarrow \mathbf{A}_i$ and $\mathbf{B}_i \leftarrow \mathbf{B}_i$ leaves $\mathbf{A}_i$ and $\mathbf{B}_i$ unchanged across clients, maintaining their pre-aggregation values.

| Method | CoLA Mcc ↑ | RTE Acc ↑ | MRPC Acc ↑ | SST-2 Acc ↑ | QNLI Acc ↑ | STS-B Corr ↑ | All Avg ↑ |
|---|---|---|---|---|---|---|---|
| Centralized LoRA$_{r=4}$ | 64.31 | 75.45 | 87.99 | 94.61 | 92.75 | 90.73 | 84.31 |
| FedIT$_{r=4}$ | 60.82 | 73.64 | 88.48 | 94.61 | 92.07 | 90.91 | 83.42 |
| FFA-LoRA$_{r=4}$ | 59.34 | 70.04 | 87.50 | 94.27 | 91.37 | 90.26 | 82.13 |
| FedEx-LoRA$_{r=4}$ | **62.82** | **75.09** | **89.95** | **94.84** | **92.66** | **90.95** | **84.39** |
| Centralized LoRA$_{r=1}$ | 62.13 | 74.67 | 87.75 | 94.61 | 92.31 | 90.83 | 83.72 |
| FedIT$_{r=1}$ | 61.33 | 71.48 | 87.99 | 94.52 | 92.01 | 90.81 | 83.02 |
| FFA-LoRA$_{r=1}$ | 57.52 | 71.20 | 87.48 | 94.03 | 91.78 | 90.34 | 82.06 |
| FedEx-LoRA$_{r=1}$ | **62.07** | **73.65** | **88.73** | **94.84** | **92.21** | **90.87** | **83.73** |

(a) Results with RoBERTa-base on the GLUE benchmark datasets

| Method | CoLA Mcc ↑ | RTE Acc ↑ | MRPC F1 ↑ | SST-2 Acc ↑ | QNLI Acc ↑ | STS-B Corr ↑ | All Avg ↑ |
|---|---|---|---|---|---|---|---|
| Centralized LoRA$_{r=4}$ | 66.03 | 82.67 | 88.84 | 96.21 | 94.58 | 91.92 | 86.71 |
| FedIT$_{r=4}$ | 64.48 | 78.43 | 88.48 | 95.87 | 94.41 | 91.29 | 85.49 |
| FFA-LoRA$_{r=4}$ | 62.05 | 75.39 | 86.52 | 95.27 | 94.35 | 90.23 | 83.97 |
| FedEx-LoRA$_{r=4}$ | **65.29** | **80.31** | **89.95** | **96.21** | **94.71** | **91.85** | **86.39** |
| Centralized LoRA$_{r=1}$ | 65.21 | 83.39 | 92.44 | 96.10 | 94.42 | 92.12 | 87.28 |
| FedIT$_{r=1}$ | 62.82 | 78.11 | 91.29 | 96.10 | 94.35 | 91.62 | 85.72 |
| FFA-LoRA$_{r=1}$ | 60.58 | 74.67 | 89.47 | 95.58 | 94.01 | 91.34 | 84.28 |
| FedEx-LoRA$_{r=1}$ | **64.35** | **80.01** | **91.76** | **96.22** | **94.71** | **91.91** | **86.49** |

(b) Results with RoBERTa-large on the GLUE benchmark datasets

Table 3: Results with RoBERTa-base and Roberta-large on the GLUE benchmark datasets, comparing various federated LoRA methods at ranks $r = \{4, 1\}$. **Centralized LoRA (in grey) sets the benchmark skyline** for its federated versions. Best results among federated methods (in blue) are highlighted in **bold** for each setting. There are 3 local epochs before every aggregation round. We report Matthew's correlation for CoLA, Pearson correlation for STS-B, and accuracy for others. Higher is better for all metrics.

| Method | E2E NLG Challenge | | | | |
|---|---|---|---|---|---|
| | BLEU ↑ | NIST ↑ | MET ↑ | ROUGE-L ↑ | CIDEr ↑ |
| Centralized LoRA$_{r=4}$ | 68.91 | 8.73 | 46.78 | 71.29 | 2.47 |
| FedIT$_{r=4}$ | 67.60 | 8.67 | 46.30 | 68.96 | 2.41 |
| FFA-LoRA$_{r=4}$ | 66.79 | 8.61 | 45.24 | 67.98 | 2.39 |
| FedEx-LoRA$_{r=4}$ | **68.15** | **8.72** | **46.48** | **69.49** | **2.44** |
| Centralized LoRA$_{r=1}$ | 67.41 | 8.68 | 46.01 | 69.51 | 2.41 |
| FedIT$_{r=1}$ | 66.01 | 8.56 | 45.21 | 68.14 | 2.28 |
| FFA-LoRA$_{r=4}$ | 65.87 | 8.54 | 45.02 | 68.05 | 2.27 |
| FedEx-LoRA$_{r=1}$ | **67.02** | **8.61** | **45.99** | **69.52** | **2.38** |

Table 4: Results with GPT-2 on the E2E NLG Challenge, comparing various federated LoRA methods at ranks $r = \{4, 1\}$. **Centralized LoRA (in grey) sets the benchmark skyline** for its federated versions. Best results among federated methods (in blue) are highlighted in **bold** for each setting. There are 3 local epochs before every aggregation round. Higher is better for all metrics.

(3) **FedEx-LoRA** aggregates $\mathbf{A}_i$ and $\mathbf{B}_i$ using the aggregation method in FedIT (FedAvg), providing the best low-rank approximation to the aggregated update with the residual $\mathbf{\Delta W}_{res}$ stored in $\mathbf{W}_0$. We present results for RoBERTa-base on the GLUE benchmark in Table 5. FedEx-LoRA outperforms the other strategies, leading us to adopt $\mathbf{B}_i \leftarrow \frac{1}{k} \sum_{i=1}^{k} \mathbf{B}_i$ and $\mathbf{A}_i \leftarrow \frac{1}{k} \sum_{i=1}^{k} \mathbf{A}_i$ across all clients.

| Method | CoLA Mcc ↑ | RTE Acc ↑ | MRPC Acc ↑ | SST-2 Acc ↑ | QNLI Acc ↑ | STS-B Corr ↑ | All Avg ↑ |
|---|---|---|---|---|---|---|---|
| Reinitialize $\mathbf{A}_i$ and $\mathbf{B}_i$ | 0.00 | 61.37 | 75.74 | 76.26 | 53.98 | 53.38 | 53.46 |
| $\mathbf{A}_i \leftarrow \mathbf{A}_i$ and $\mathbf{B}_i \leftarrow \mathbf{B}_i$ | 55.54 | 59.93 | 84.80 | 92.77 | 88.98 | 88.41 | 78.41 |
| FedEx-LoRA | **62.82** | **75.09** | **89.95** | **94.84** | **92.66** | **90.95** | **84.39** |

Table 5: Results with RoBERTa-base ($r = 4$) on the GLUE benchmark datasets, comparing various assignment strategies for $\mathbf{A}_i$ and $\mathbf{B}_i$. We report Matthew's correlation for CoLA, Pearson correlation for STS-B, and accuracy for other datasets. Best results for each dataset are highlighted in **bold**.

To extend our method to rank-heterogeneous settings, the assignments for $\mathbf{A}_i$ and $\mathbf{B}_i$ must also accommodate rank heterogeneity. Further investigation is required to develop an optimal assignment strategy that supports this.

**Scaled Frobenius Norm of Divergence/Deviation.** We now study the deviations in updates from federated averaging (FedAvg) relative to ideal updates and analyze the findings. To quantify this deviation, we measure the scaled Frobenius norm of the divergence between the updates produced by FedAvg and the ideal LoRA updates, revealing several notable patterns. In Figure 2, we plot this divergence for the query (Q) and value (V) matrices across model layers, computed after the first aggregation step for local epochs = $\{3, 10\}$. We observe that (1) the deviations decrease as the model depth increases, (2) the deviation grows with a higher number of local epochs, and (3) the deviation is more pronounced in the query (Q) matrices compared to the value (V) matrices. These trends hold consistently across various datasets and settings, as shown by additional plots in Appendix F.1 (see Figures 4 and 5).

Next, we examine how this deviation evolves across multiple rounds of federated aggregation. We plot the scaled Frobenius norm of the deviation between FedAvg and ideal LoRA updates over several aggregation rounds for different datasets, focusing on (a) the query matrices of the first layer, and (b) the average of the query and value matrices across all layers, as presented in Figure 3. We observe that the deviation consistently decreases as the number of aggregation rounds increases, both for the first-layer query matrix and for the average of the query and value matrices across all layers. These findings are further supported by detailed plots across multiple datasets and settings, as shown in Appendix F.2 (see Figures 6, 7, 8, and 9).

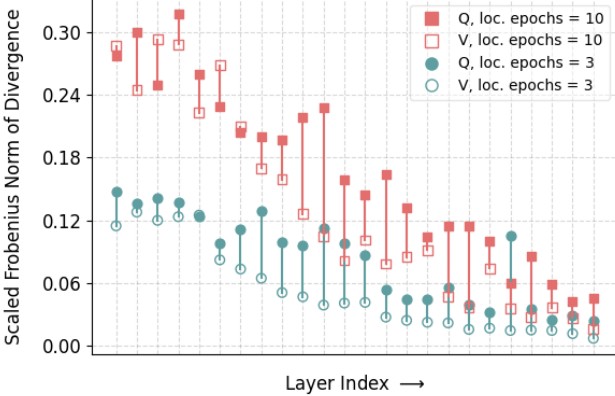

Figure 2: Scaled Frobenius norm of divergence/deviation of updates with conventional federated aggregation (FedAvg) versus ideal LoRA updates, computed after the first aggregation step. We plot for query (Q) and value (V) matrices across model layers. Results are shown for local epochs = $\{3, 10\}$. (Dataset: MRPC, model: RoBERTa-large, $r = 1$).

**Communication Costs.** As discussed in Section 4, FedEx-LoRA transmits a higher-rank update matrix (rank = $k \cdot r$) along with the low-rank adapters, which raises concerns about potential communication overhead. Table 6 compares the communication costs of FFA-LoRA, FedIT, and

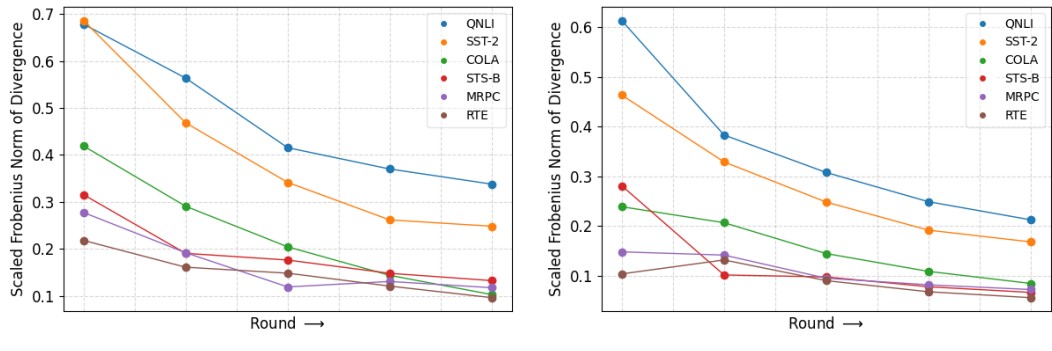

(a) Query matrices of first layer          (b) Avg. of query and value matrices across all layers

Figure 3: Scaled Frobenius norm of divergence/deviation of updates with conventional federated aggregation (FedAvg) versus ideal LoRA updates, computed across multiple aggregation rounds for various datasets. We present results for (a) query matrices from the first layer, and (b) the average of query and value matrices across all layers. (Model: RoBERTa-large, $r = 1$, local epochs $= 10$).

full federated fine-tuning (FT), compared to FedEx-LoRA, for RoBERTa-base, RoBERTa-large, and GPT-2 models with rank $r = 4$ over 5 communication rounds. FedEx-LoRA incurs only a marginal increase in communication overhead relative to FedIT and FFA-LoRA, while FFA-LoRA has the lowest cost due to its reduced number of trainable parameters. FedEx-LoRA still maintains a substantially lower communication cost compared to federated full FT.

The practical impact of communication overhead is reduced by two factors: **(1)** the initial transmission of full model weights dominates communication costs, and **(2)** in NLU tasks, most communicated parameters come from the classification head, which requires training regardless of the aggregation method. Therefore, communication cost differences between FedEx-LoRA, FedIT, and FFA-LoRA are minimal in practice. Despite this marginal overhead, FedEx-LoRA consistently outperforms other federated LoRA approaches, making it an effective choice for federated fine-tuning.

| Model | Federated Full FT | FedEx-LoRA | FedIT | FFA-LoRA |
|---|---|---|---|---|
| RoBERTa-base | 7.032 | 1 | 0.979 | 0.972 |
| RoBERTa-large | 10.396 | 1 | 0.984 | 0.979 |
| GPT-2 | 9.475 | 1 | 0.917 | 0.886 |

Table 6: Ratio of # of parameters communicated in federated LoRA variants and federated full FT to FedEx-LoRA. All results are reported with rank $r = 4$ and across 5 communication rounds.

## 7 CONCLUSION

In our work, we identified limitations in state-of-the-art federated fine-tuning methods that struggle with inexact aggregation. We proposed a novel method, FedEx-LoRA, which appends the residual error matrix to the frozen pretrained matrix, while maintaining minimal communication and computational overhead. The strength of our approach lies in its simplicity and broad applicability. Extensive experiments demonstrate that FedEx-LoRA consistently outperforms other federated LoRA methods across various datasets and settings. Our analyses reveal that deviations in updates from federated averaging compared to the ideal solution are significant and exhibit notable patterns.

Testing in privacy-preserving scenarios is a natural extension of our work. FFA-LoRA (Sun et al., 2024) demonstrated that noise in differential privacy leads to greater deviations from ideal updates. Given that our method achieves exact aggregation and outperforms FFA-LoRA in non-private settings, we anticipate similar success in privacy-sensitive applications. Our approach can be readily adapted for fine-tuning other models like Vision Transformers (ViTs) and Vision-Language models (VLMs).

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

## A  DATASET DETAILS

**COMMONSENSE170K** is a dataset combining eight commonsense reasoning datasets (Hu et al., 2023), as detailed below:

1. **WinoGrande** (Sakaguchi et al., 2021) involves filling in blanks with binary choices based on sentences that demand commonsense reasoning.

2. **HellaSwag** (Zellers et al., 2019) asks the model to predict the most plausible continuation of a given context by selecting the correct ending from several options.

3. **ARC Challenge** or **ARC-c** (Clark et al., 2018) consists of multiple-choice science questions designed to challenge models with more complex reasoning, making them harder for methods that rely solely on co-occurrence patterns.

4. **PIQA** (Bisk et al., 2020) tests physical commonsense reasoning, where the task is to choose the best action from a set of options in a hypothetical situation.

5. **BoolQ** (Clark et al., 2019) focuses on yes/no question answering from naturally occurring queries.

6. **ARC Easy** or **ARC-e** (Clark et al., 2018) consists of grade-school-level multiple-choice science questions, providing a simpler set of tasks for testing models' basic reasoning abilities.

7. **OBQA** (Mihaylov et al., 2018) contains open-book, knowledge-intensive QA tasks requiring multi-hop reasoning to answer questions that involve integrating information from multiple sources.

8. **SIQA** (Sap et al., 2019) focuses on understanding human actions and predicting their social consequences, evaluating models' social commonsense reasoning.

**MetaMathQA** dataset (Yu et al., 2024) generates mathematical questions by rephrasing them from various perspectives without introducing additional knowledge. We evaluate this dataset on two benchmarks: GSM8K (Cobbe et al., 2021), which includes grade-school math word problems that require multi-step reasoning, and MATH (Hendrycks et al., 2021), which features challenging competition-level mathematics problems.

**GLUE Benchmark** is a diverse suite of tasks for evaluating natural language understanding capabilities. It includes datasets such as SST-2 for sentiment analysis (Socher et al., 2013), MRPC for paraphrase detection (Dolan & Brockett, 2005), CoLA for linguistic acceptability (Warstadt et al., 2019), QNLI for inference (Rajpurkar et al., 2018), RTE for inference, and STS-B for semantic textual similarity (Cer et al., 2017). Due to its comprehensive coverage of NLU tasks, GLUE is widely used to assess models like RoBERTa. Each dataset is released under its own license.

The **E2E NLG Challenge** (Novikova et al., 2017) dataset is widely used to evaluate systems for natural language generation, particularly for data-to-text tasks. It contains around 42,000 training examples, with an additional 4,600 each for validation and testing, all from the restaurant domain. Each input table has multiple reference outputs, where each data point $(x, y)$ includes a sequence of slot-value pairs and its corresponding reference text in natural language. The dataset is made available under the Creative Commons BY-NC-SA 4.0 license.

## B  HYPERPARAMETER DETAILS

We conduct experiments on a single NVIDIA A100/A6000 GPU and report the average results from three independent runs. All models are trained using the AdamW optimizer (Loshchilov & Hutter, 2019). For the instruction tuning experiments, the hyperparameters and configurations for Mistral-7B, Gemma-2 9B, and Llama-3.2 3B are provided in Table 7, following most of the settings from previous works (Hu et al., 2023). The hyperparameter configurations for GPT-2 and RoBERTa-base/large are detailed in Table 8, with most settings following the original LoRA paper (Hu et al., 2021), except for a learning rate sweep.

|  | **Mistral-7B / Gemma-2 9B** | **Llama-3.2 3B** |
|---|---|---|
| Optimizer | AdamW | AdamW |
| Batch size | 1 | 6 |
| Max. Seq. Len | 512 | 256 |
| Grad Acc. Steps | 32 | 24 |
| Local Epochs | 1 | 1 |
| Rounds | 1 | 1 |
| Dropout | 0 | 0 |
| Learning Rate | $5e-4$ | $5e-4$ |
| LR Scheduler | Cosine | Linear |
| Warmup Ratio | 0.02 | 0.02 |
| LoRA $\alpha$ | 16 | 16 |

Table 7: Hyperparameter settings for Mistral-7B, Gemma-2 9B & Llama-3.2 3B.

|  | **GPT-2** | **RoBERTa-base/large** |
|---|---|---|
|  | Training | |
| Optimizer | AdamW | AdamW |
| Weight Decay | 0.01 | 0.01 |
| Dropout Prob | 0.1 | 0.1 |
| Batch Size | 8 | 128 |
| Warmup Steps | 500 | - |
| Warmup Ratio | - | 0.6 |
| Label Smooth | 0.1 | - |
| Max Seq. Len | 128 | 512 |
| Learning Rate | $2 \cdot 10^{-3}$ | $1 \cdot 10^{-3}$ |
| LoRA $\alpha$ | 32 | 8 |
|  | Inference | |
| Beam Size | 10 | - |
| Length Penalty | 0.9 | - |
| no repeat ngram size | 4 | - |

Table 8: Hyperparameter settings for GPT-2 and RoBERTa-base/large.

## C EFFECT OF VARYING RANK

We evaluate FedEx-LoRA against other federated fine-tuning methods on the CoLA dataset using RoBERTa-base, by varying the rank of the low-rank adapters across $r = \{1, 2, 4, 8, 16, 32\}$, as presented in Table 9. Across all rank configurations, FedEx-LoRA consistently outperforms competing federated LoRA variants. In agreement with prior studies (Hu et al., 2021; Zhang et al., 2023b), increasing the rank does not always result in performance gains. For this task, we find that the optimal performance is achieved at $r = 8$, beyond which further increases in rank yield diminishing returns.

| Method | r = 1 | r = 2 | r = 4 | r = 8 | r = 16 | r = 32 |
|---|---|---|---|---|---|---|
| Centralized LoRA | 62.13 | 62.11 | 64.31 | 64.44 | 64.32 | 63.98 |
| FedIT | 60.05 | 60.32 | 60.82 | 62.09 | 62.15 | 61.98 |
| FFA-LoRA | 57.73 | 57.78 | 59.34 | 57.82 | 57.78 | 58.24 |
| FedEx-LoRA | **62.07** | **61.38** | **62.82** | **63.57** | **63.56** | **63.35** |

Table 9: Matthew's correlation on CoLA across different ranks for various federated LoRA methods. **Centralized LoRA (in grey) sets the benchmark skyline** for its federated versions. Best results among federated methods (in blue) are highlighted in **bold** for each rank. (Model: RoBERTa-base, local epochs = 3).

# D ADDITIONAL EXPERIMENTS FOR NLU

We present additonal results with the RoBERTa-base and RoBERTa-large models in Table 10, evaluated at ranks $r = \{4, 1\}$, with local epochs set to 10.

| Method | CoLA Mcc ↑ | RTE Acc ↑ | MRPC Acc ↑ | SST-2 Acc ↑ | QNLI Acc ↑ | STS-B Corr ↑ | All Avg ↑ |
|---|---|---|---|---|---|---|---|
| Centralized LoRA$_{r=4}$ | 64.31 | 75.45 | 87.99 | 94.61 | 92.75 | 90.73 | 84.31 |
| FedIT$_{r=4}$ | 58.55 | 70.75 | 87.50 | 94.36 | 92.09 | 90.58 | 82.31 |
| FFA-LoRA$_{r=4}$ | 57.52 | 71.84 | 86.76 | 94.24 | 91.27 | 90.04 | 81.95 |
| FedEx-LoRA$_{r=4}$ | **61.32** | **75.81** | **87.75** | **94.57** | **92.64** | **90.62** | **83.79** |
| Centralized LoRA$_{r=1}$ | 62.13 | 74.67 | 87.75 | 94.61 | 92.31 | 90.83 | 83.72 |
| FedIT$_{r=1}$ | 60.05 | 71.84 | 88.79 | 94.62 | 92.23 | 90.54 | 83.01 |
| FFA-LoRA$_{r=1}$ | 57.73 | 71.18 | 87.74 | 93.69 | 91.41 | 90.18 | 81.99 |
| FedEx-LoRA$_{r=1}$ | **61.31** | **73.12** | **89.21** | **94.73** | **92.40** | **90.67** | **83.57** |

(a) Results with RoBERTa-base on the GLUE benchmark datasets

| Method | CoLA Mcc ↑ | RTE Acc ↑ | MRPC F1 ↑ | SST-2 Acc ↑ | QNLI Acc ↑ | STS-B Corr ↑ | All Avg ↑ |
|---|---|---|---|---|---|---|---|
| Centralized LoRA$_{r=4}$ | 66.03 | 82.67 | 88.84 | 96.21 | 94.58 | 91.92 | 86.71 |
| FedIT$_{r=4}$ | 61.80 | 77.83 | 85.54 | 95.83 | 94.32 | 91.70 | 84.50 |
| FFA-LoRA$_{r=4}$ | 60.16 | 74.67 | 84.31 | 95.64 | 94.29 | 90.28 | 83.23 |
| FedEx-LoRA$_{r=4}$ | **62.60** | **79.19** | **86.03** | **96.10** | **94.74** | **91.91** | **85.10** |
| Centralized LoRA$_{r=1}$ | 65.21 | 83.39 | 89.21 | 96.10 | 94.42 | 92.12 | 86.74 |
| FedIT$_{r=1}$ | 61.06 | 78.33 | 88.48 | 95.86 | 94.25 | 91.17 | 84.85 |
| FFA-LoRA$_{r=1}$ | 60.32 | 72.45 | 85.78 | 95.52 | 93.94 | 91.25 | 83.21 |
| FedEx-LoRA$_{r=1}$ | **63.56** | **79.07** | **89.71** | **96.22** | **94.56** | **91.77** | **85.82** |

(b) Results with RoBERTa-large on the GLUE benchmark datasets

Table 10: Results with RoBERTa-base and Roberta-large on the GLUE benchmark datasets, comparing various federated LoRA methods at ranks $r = \{4, 1\}$. There are 10 local epochs before every aggregation round.

# E ADDITIONAL EXPERIMENTS FOR NLG

Table 11 presents additional experiments of GPT-2 fine-tuned with ranks $r = \{4, 1\}$, with local epochs set to 5. FedEx-LoRA consistently outperforms leading federated fine-tuning methods across all metrics and settings, consistent with the results presented in Table 4.

| Method | E2E NLG Challenge | | | | |
|---|---|---|---|---|---|
| | BLEU ↑ | NIST ↑ | MET ↑ | ROUGE-L ↑ | CIDEr ↑ |
| Centralized LoRA$_{r=4}$ | 68.91 | 8.73 | 46.78 | 71.29 | 2.47 |
| FedIT$_{r=4}$ | 67.61 | 8.62 | 46.45 | 70.28 | 2.43 |
| FFA-LoRA$_{r=4}$ | 67.21 | 8.57 | 46.05 | 69.98 | 2.41 |
| Exact-FedIT$_{r=4}$ | **68.49** | **8.72** | **46.76** | **70.71** | **2.48** |
| Centralized LoRA$_{r=1}$ | 67.41 | 8.68 | 46.01 | 69.51 | 2.41 |
| FedIT$_{r=1}$ | 66.16 | 8.56 | 45.54 | 68.25 | 2.29 |
| FFA-LoRA$_{r=1}$ | 65.78 | 8.49 | 45.01 | 67.82 | 2.26 |
| Exact-FedIT$_{r=1}$ | **66.54** | **8.57** | **46.07** | **69.11** | **2.37** |

Table 11: Results with GPT-2 on the E2E NLG Challenge, comparing various federated LoRA methods at ranks $r = \{4, 1\}$. There are 5 local epochs before every aggregation round.

# F MORE DIVERGENCE/DEVIATION PLOTS

## F.1 DEVIATION/DIVERGENCE PLOTS ACROSS LAYERS

As discussed in Section 6, we further quantify the deviation of conventional federated aggregation (FedAvg) from ideal updates by measuring the scaled Frobenius norm of the divergence the updates produced by FedAvg and the ideal LoRA updates. We present additional plots of this divergence for the query (Q) and value (V) matrices across model layers, computed after the first aggregation step for local epochs $= \{3, 10\}$ across multiple datasets, in Figures 4 and 5. Figure 4 shows results for rank $r = 1$, while Figure 5 presents results for rank $r = 4$.

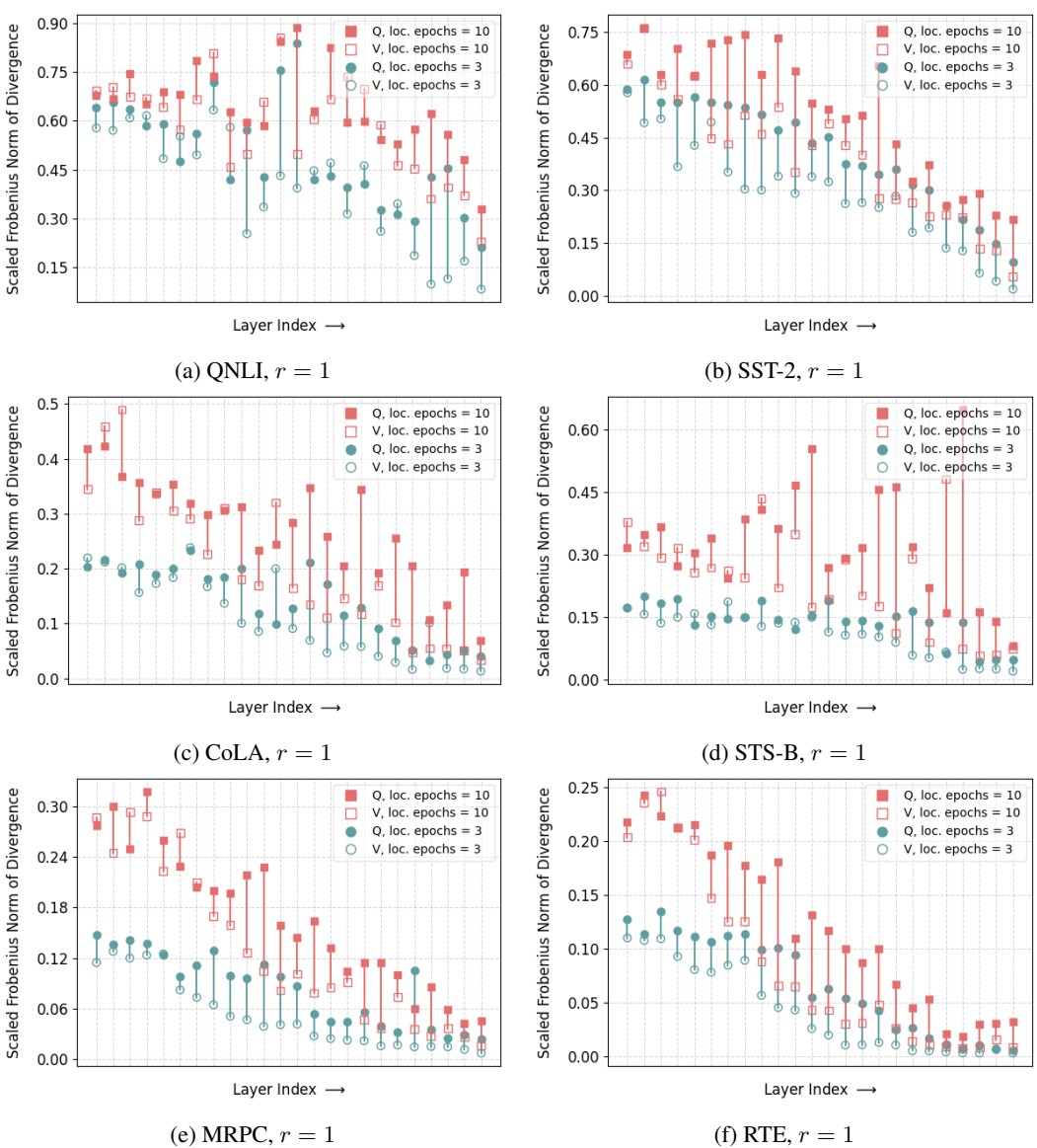

Figure 4: Scaled Frobenius norm of divergence/deviation of updates with conventional federated aggregation (FedAvg) versus ideal LoRA updates, computed after the first aggregation step. We plot for query (Q) and value (V) matrices across model layers, for multiple datasets. Results are shown for local epochs $= \{3, 10\}$. (Model: RoBERTa-large, $r = 1$).

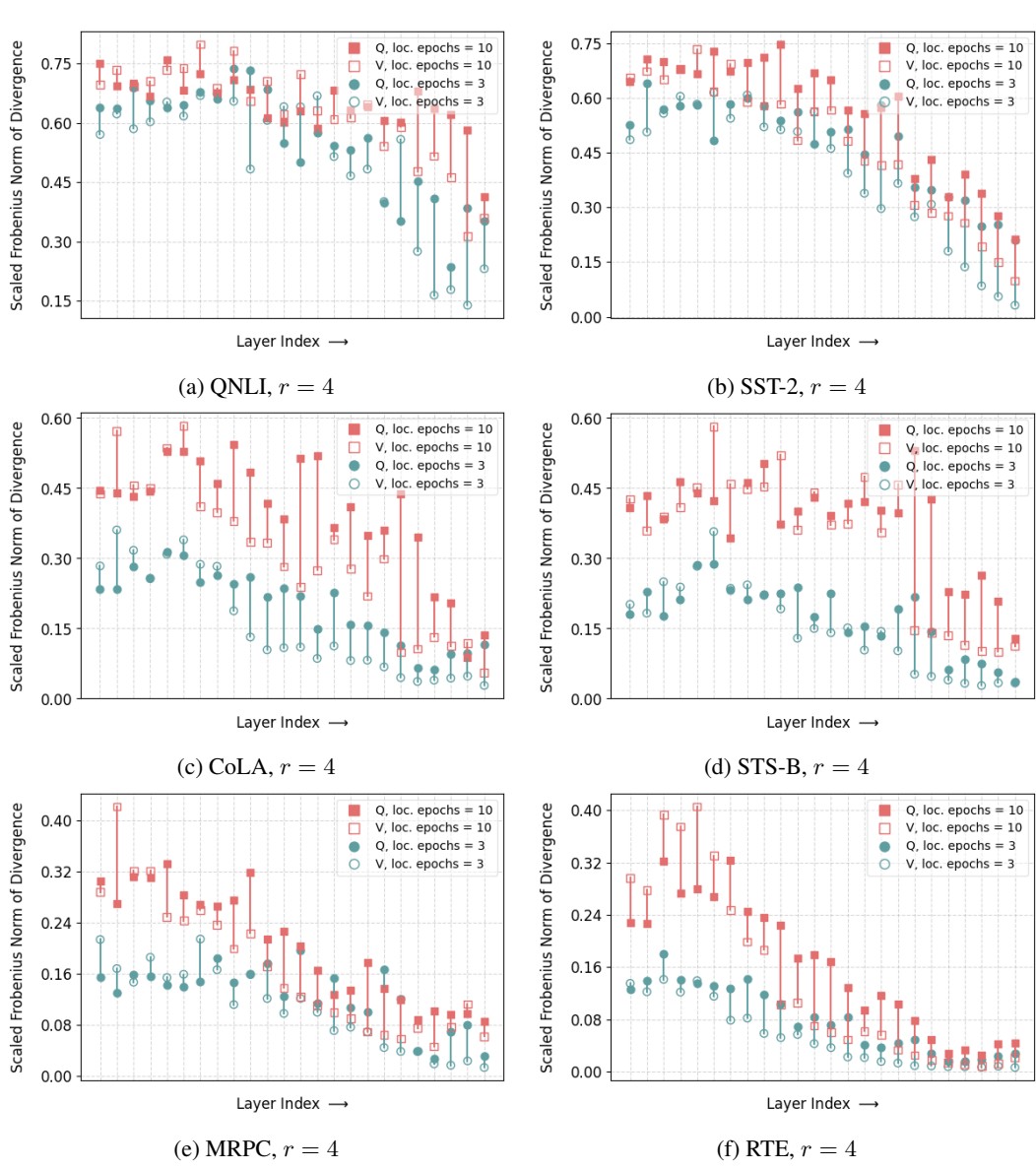

Figure 5: Scaled Frobenius norm of divergence/deviation of updates with conventional federated aggregation (FedAvg) versus ideal LoRA updates, computed after the first aggregation step. We plot for query (Q) and value (V) matrices across model layers, for multiple datasets. Results are shown for local epochs $= \{3, 10\}$. (Model: RoBERTa-large, $r = 4$).

We now examine how the deviation evolves across multiple rounds of federated aggregation. We plot the scaled Frobenius norm of the deviation between FedAvg and ideal LoRA updates over several aggregation rounds for different datasets, focusing on (a) the query matrices of the first layer and (b) the average of the query and value matrices across all layers. This is presented in Figures 6, 7, 8, and 9. We include results for ranks $r = \{1, 4\}$ and local epochs $= \{3, 10\}$.

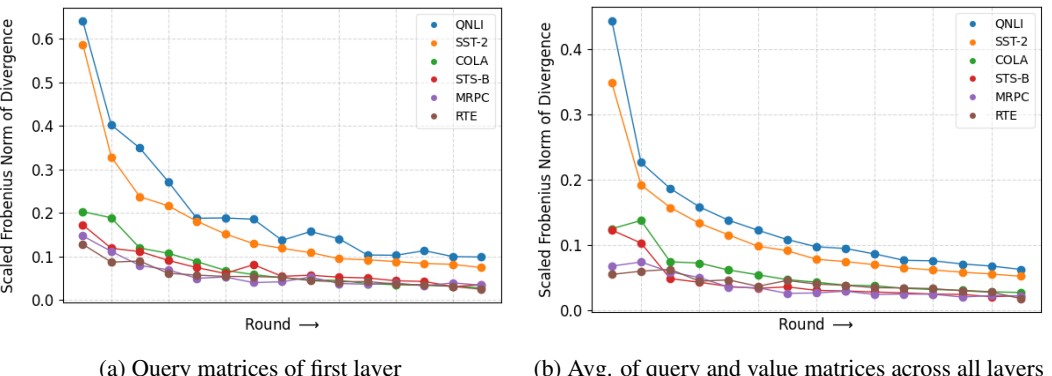

(a) Query matrices of first layer    (b) Avg. of query and value matrices across all layers

Figure 6: Scaled Frobenius norm of divergence/deviation of updates with conventional federated aggregation (FedAvg) versus ideal LoRA updates, computed across multiple aggregation rounds for various datasets. We present results for (a) query matrices from the first layer, and (b) the average of query and value matrices across all layers. (Model: RoBERTa-large, $r = 1$, local epochs = 3)

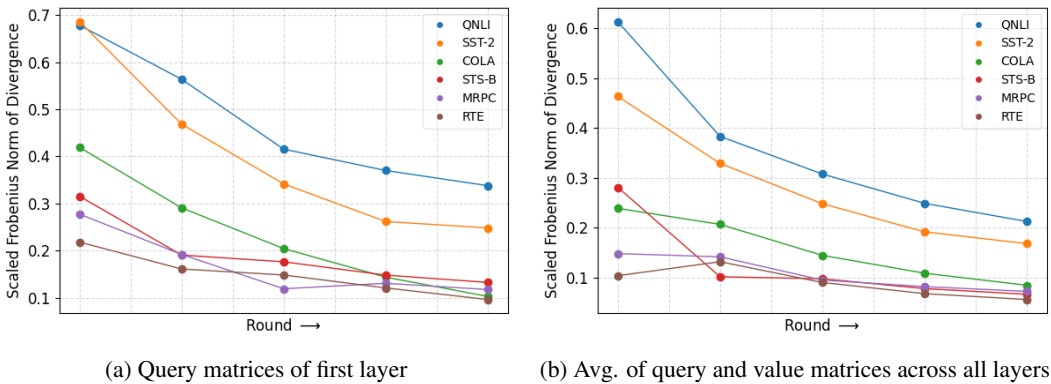

(a) Query matrices of first layer    (b) Avg. of query and value matrices across all layers

Figure 7: Scaled Frobenius norm of divergence/deviation of updates with conventional federated aggregation (FedAvg) versus ideal LoRA updates, computed across multiple aggregation rounds for various datasets. We present results for (a) query matrices from the first layer, and (b) the average of query and value matrices across all layers. (Model: RoBERTa-large, $r = 1$, local epochs = 10)

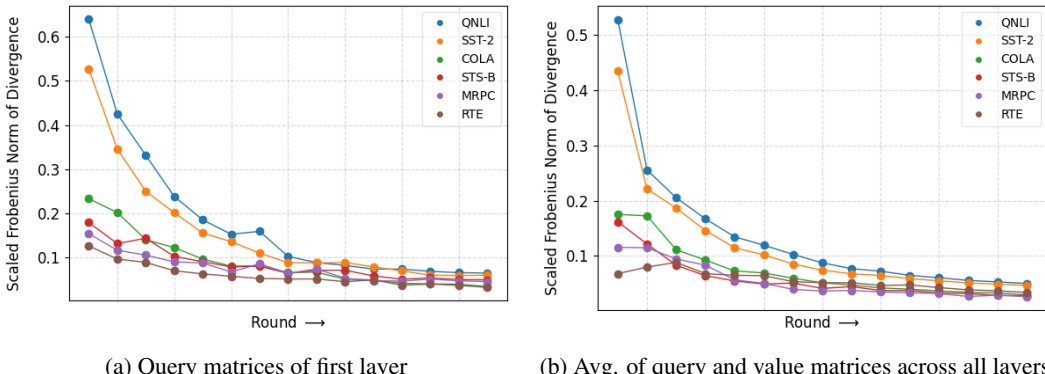

(a) Query matrices of first layer  (b) Avg. of query and value matrices across all layers

Figure 8: Scaled Frobenius norm of divergence/deviation of updates with conventional federated aggregation (FedAvg) versus ideal LoRA updates, computed across multiple aggregation rounds for various datasets. We present results for (a) query matrices from the first layer, and (b) the average of query and value matrices across all layers. (Model: RoBERTa-large, $r = 4$, local epochs = 3)

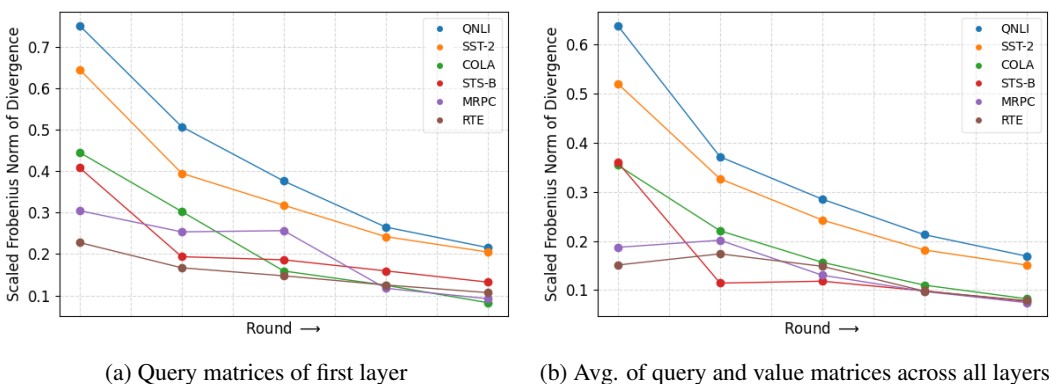

(a) Query matrices of first layer  (b) Avg. of query and value matrices across all layers

Figure 9: Scaled Frobenius norm of divergence/deviation of updates with conventional federated aggregation (FedAvg) versus ideal LoRA updates, computed across multiple aggregation rounds for various datasets. We present results for (a) query matrices from the first layer, and (b) the average of query and value matrices across all layers. (Model: RoBERTa-large, $r = 4$, local epochs = 10)

