# OpenReview forum: "FedEx-LoRA: Exact Aggregation for Federated and Efficient Fine-Tuning of Foundation Models"
_ICLR.cc/2025/Conference — ICLR 2025 Conference Withdrawn Submission_

### Official Review · Reviewer_XWEi · 2024-10-24

**Soundness:** 2
**Presentation:** 1
**Contribution:** 1
**Rating:** 3
**Confidence:** 5

**Summary:**

Given the problem that averages the A and B matrices of LoRA will result in inexact updates in federated learning, this paper proposes Federated Exact LoRA (FedEx-LoRA). FedEx-LoRA computes the aggregation error on the server, sends the error weights to the clients, and updates the local model by the weights while preserving the aggregated LoRA. The paper provides experiments on RoBERTa and GPT-2 to demonstrate the effectiveness of FedEx-LoRA compared to FedIT and FFA-LoRA.

**Strengths:**

(1) The problem raised about whether to use FedAvg on LoRA is valuable. It brings up the problem that the optimization target of LoRA federated fine-tuning is the full global model $W$ or a global LoRA $B_GA_G$.

(2) Figure 2 provides some interesting insights for FL researchers. The authors find that the deviations decrease as the model depth increases, which may lead to deeper reflections on the relationship between model architecture and fine-tuning.

**Weaknesses:**

**Contribution**: The authors mention in the paper that they identify a critical discrepancy in traditional federated averaging of LoRA adapters. However, this question has been raised and studied in detail. [1] and [2] identified this problem and [2] provided a solution for eliminating this error. None of these previous works are discussed in detail and listed as baselines. The reviewer believes that the author has not adequately researched this field, so the contribution is limited.

**Method**: The reviewer believes that the solution proposed in this paper is too simplistic and impractical. Here are some reasons:

(1) Increased communication cost: According to Figure 1 and Eq. (14), the server needs to send the residual error term to clients for updating the local base model. This makes the communication cost from server to clients as large as full fine-tuning (or at least the full fine-tuning on the layers that apply LoRA). It is not acceptable since we use LoRA to reduce the communication cost. Especially in federated fine-tuning LLMs, this residual error term may be as large as $kT$ times the LLM parameter size.

(2) Increased computation cost: According to Figure 1 and Eq. (11) (12), the server will iteratively compute the production of $BA$ from clients and the residual error term. This has led to a several-fold increase in computation on the server, compared to FedIT.

**Writing**: This paper has some typos, as well as unclear parts that do not conform to the federated learning paradigm.

(1) Eq. (5), (7) look confused, why the aggregation of local LoRA modules $B_i$ and $A_i$ is still the local LoRA? If here the authors mean that the new $B_i A_i$ are local modules in the next epoch, why in Eq. (7) the global model is updated by the local LoRA? These expressions may be understandable to researchers familiar with federated learning through guessing, but they are still difficult to understand.

(2) The paper use $B \cdot A$ in Figure 1. It seems incorrect to use the dot product here. However, in the equations, $BA$ and $B\times A$ are used.

(3) $W_0$, $W^{global}$, $A_{init} B_{init}$, $A_{i} B_{i}$ are used together. It is confusing which round these weights are in and whether they are the global or local weights. It could be better to use communication round numbers and uniform global and local marks.

**Experiments**: Related papers such as FedIT are using LLMs for experiments. But FedEx-LoRA is using RoBERTa and GPT-2. It is understandable that experimental resources are limited, but the research related to LoRA nowadays lacks sufficient convincing power when applied to models smaller than 1b parameters. As mentioned before, the extra communication and computation costs are not well analyzed.

**Missed Baselines**: The paper misses detailed comparison to at least two important related works that focus on the same topic:

[1] Sun, Y., Li, Z., Li, Y., & Ding, B. (2024). Improving loRA in privacy-preserving federated learning. arXiv preprint arXiv:2403.12313.

[2] Wang, Z., Shen, Z., He, Y., Sun, G., Wang, H., Lyu, L., & Li, A. (2024). Flora: Federated fine-tuning large language models with heterogeneous low-rank adaptations. arXiv preprint arXiv:2409.05976.

**Questions:**

Please see the weakness.

---

> ### Author Response · Authors · 2024-11-20
> **Reviewer XWEi Response**
>
> **Contribution/Missed Baselines**
>
> We appreciate the reviewer highlighting these works.
>
> The paper "Improving LoRA in Privacy-Preserving Federated Learning" [1], referred to as FFA-LoRA in their work, was thoroughly addressed in our original manuscript.  We explicitly discussed it in Section 3 (lines 170-176), included it as a baseline (lines 269-271), and provided detailed comparisons across all experiments. In the original manuscript, these comparisons appear in Tables 1, 2, 5, and 7, while in the new manuscript, they are found in Tables 1, 2, 3, 4, 9, 10, and 11. We are unsure how the reviewer missed these multiple mentions of FFA-LoRA in our original paper.
>
> Our analysis covered its trade-offs between exactness and expressivity, and evaluated it on multiple settings. Our findings highlighted the impact of its choice to freeze the A matrices, as we consistently outperformed FFA-LoRA across all models and datasets.
>
> Regarding FLORA [2], *per ICLR 2025 guidelines, papers published after July 1, 2024 are considered contemporaneous work. As FLORA was released in September 2024, while relevant, it falls outside the required comparison window for submissions.
> *Here is the link for ICLR's explicit policy: (https://iclr.cc/Conferences/2025/FAQ)
>
>
> **Experiments**
>
> We appreciate these suggestions and have addressed these points in our revision. We have expanded our work to evaluate larger models and more complex tasks. Our method **significantly** outperforms state-of-the-art federated LoRA approaches when evaluated on complex tasks with LLMs. This improvement is much more pronounced compared to our previous evaluations with smaller models. The manuscript now includes detailed evaluations of models ranging from 3B to 9B parameters, covering both arithmetic and commonsense reasoning tasks. For arithmetic reasoning, we fine-tuned Mistral-7B and Gemma-2 9B on a subset of the MetaMathQA dataset and evaluated their performance using the GSM8K and MATH benchmarks. For commonsense reasoning, we fine-tuned Llama-3.2 3B on COMMONSENSE170K—a collection of eight commonsense reasoning datasets—and reported performance for each task. The results, presented in Tables 1 and 2 (of the revised manuscript), demonstrate that our method consistently outperforms other methods across all tasks.
>
>  **Method**
>
> **Communication Cost**
>
> The communication overhead is carefully managed through matrix decomposition. Specifically, since ∆Wres has rank ≤ (kr), we can decompose it using Singular Value Decomposition (SVD) into U, S, V. Now the singular values will be 0 post row and column > (kr). We can simply take the first (kr) rows of U and similar columns for V. This decomposition means we only need to transmit matrices of size m×(kr) and (kr)×n, requiring (kr)(m+n) parameters. In contrast, transmitting the full weight update would require m×n parameters.
>
> For concrete quantification, consider our RoBERTa-base experiments where m=n=768, k=3 clients, and r=4: FedEx-LoRA requires transmitting only 2×768×(3×4) = 18,432 parameters per layer, compared to 768×768 = 589,824 parameters for full model updates. This represents a 32x reduction in communication cost per layer (the total cost is proportional to the cost per layer). This analysis aligns with our empirical measurements in Table 4 (original manuscript) or Table 6 (revised manuscript), where FedEx-LoRA shows only marginal overhead compared to standard federated LoRA approaches while being significantly cheaper than federated full fine-tuning.
>
>
> **Computation Cost**
>
> The matrix multiplications required for aggregation are minimal compared to the computational demands of model training on clients. For perspective, a single forward pass by any client requires orders of magnitude more computation than the server's aggregation operations. The training computations dwarf the simple matrix operations needed for aggregation.
> Recent work in federated learning (Kairouz, et al.,2021) advocates for leveraging server computation to improve system efficiency. We believe that our minimal computational overhead should not be considered a weakness as it aligns well with modern FL approaches.
>
> **Writing**
>
> We appreciate the reviewer's suggestions to enhance the clarity of the manuscript. We have revised the equations in Section 4 for consistent notation. Superscripts now indicate the aggregation round, while subscripts denote the client number. Global weights and adapters are clearly specified in the subscript for clarity.
>
>
>
> References:
>
> [1] Sun, Y., Li, Z., Li, Y., & Ding, B. (2024). Improving LoRA in privacy-preserving federated learning. arXiv preprint arXiv:2403.12313
>
>
> [2] Wang, Z., Shen, Z., He, Y., Sun, G., Wang, H., Lyu, L., & Li, A. (2024). Flora: Federated fine-tuning large language models with heterogeneous low-rank adaptations. arXiv preprint arXiv:2409.05976
>
> [3] P. Kairouz, H. B. McMahan, B. Avent, et al., "Advances and open problems in federated learning," arXiv preprint arXiv:1912.04977, 2021

---

> > ### Comment · Reviewer_XWEi · 2024-11-25
> >
> > Thanks for the rebuttal from authors. Thanks for the clarification according to the contribution.
> > Since the rebuttal according to the communication and computation cost parts lack experimental evidence and data support. I believe the authors don't actually have a solid way to address these issues, so I will maintain my score.
> > In addition, I did not see the revision regarding the figure.

---

> > > ### Author Response · Authors · 2024-11-28
> > > **Response**
> > >
> > > We thank the reviewer for their response to the updated draft and rebuttal.
> > > We present additional experimental results for our communication protocol in the hyperclient setting.
> > >
> > > We fine-tuned Mistral-7B with 100 clients on a subset of MetaMathQA and evaluated on GSM8K and MATH benchmarks - scenarios where the original method would incur prohibitive communication costs. Since communication cost scales linearly with the truncated SVD rank of the residual matrix and equals FedIT's cost at the same rank, our results directly demonstrate the efficiency-performance trade-off:
> > >
> > > | Method | Rank | GSM8K | MATH |
> > > |--------|------|--------|-------|
> > > | FedIT | 32 | 54.89 | 13.64 |
> > > | FedEx | 32 | **54.96** | **14.02** |
> > > | FedEx | 16 | 54.36 | 13.84 |
> > >
> > > These results show that our method not only outperforms FedIT at equal communication bandwidth but maintains similar performance even when using half the bandwidth. This demonstrates that our communication protocol effectively addresses the overhead concerns while improving upon baseline efficiency.
> > >
> > > Regarding computational overhead of SVD: While we acknowledge the increased computational requirement, several key points demonstrate why this is not a significant limitation:
> > >
> > > 1. The SVD computation is performed once per layer post-aggregation and crucially does not scale with the number of clients. This fixed overhead is more efficient than computing full SVD since only top vectors are needed.
> > >
> > > 2. Our approach is computationally more efficient than existing methods in federated LoRA literature:
> > >    - In [1], the server performs ADAM optimization steps, which are orders of magnitude more computationally expensive than truncated SVD per layer
> > >    - In [2], the server calculates full SVD decomposition for each client, resulting in client-linear scaling (Their Frobenius norm approach for efficient SVD can be directly adapted to our method)
> > >    - In [3], the server aggregates full finetuning matrices before SVD for adapter initialization, incurring comparable computational costs
> > >
> > > These established works in literature demonstrate that server-side computation overhead is an acceptable trade-off when it enables improved communication efficiency and performance in federated LoRA settings. Our method maintains this principle while providing better tunable communication-performance trade-offs.
> > >
> > > [1] Kuo et al. “Federated LoRA with Sparse Communication” , arXiv:2406.05233, 2024
> > >
> > > [2] Cho et al. "Heterogeneous LoRA for Federated Fine-tuning of On-Device Foundation Models", arXiv:2401.06432, 2024
> > >
> > > [3] S. Babakniya, A. Elkordy, Y. Ezzeldin, Q. Liu, K.-B. Song, M. EL-Khamy, and S. Avestimehr. SLoRA: Federated parameter efficient fine-tuning of language models. In International Workshop on Federated Learning in the Age of Foundation Models in Conjunction with NeurIPS 2023, 2023. URL https://openreview.net/forum?id=06quMTmtRV.

---

> ### Author Response · Authors · 2024-11-24
> **Reminder**
>
> Thank you for your comments and feedback. We would like to specifically highlight the point regarding comparisons to FFA-LoRA and FLoRA. FFA-LoRA was thoroughly addressed in our manuscript through discussions, baselines, and detailed comparisons across multiple experiments, as explained in our rebuttal. Regarding FLoRA, as per ICLR 2025 guidelines, it falls outside the required comparison window, being released in September 2024.
>
> We believe this clarification is critical, and we kindly request your response to this point before the discussion period ends. For the rest of your feedback, we hope our rebuttal and updated draft have addressed your questions and incorporated your suggestions. Please feel free to share any further questions or thoughts before the discussion period ends.

---

### Official Review · Reviewer_eEFJ · 2024-11-02

**Soundness:** 3
**Presentation:** 4
**Contribution:** 3
**Rating:** 6
**Confidence:** 4

**Summary:**

The paper introduces FedEx-LoRA, a novel approach for federated fine-tuning of LLMs using LoRA. Existing federated fine-tuning methods face limitations in achieving exact aggregation due to the inexact averaging of LoRA adapters. The proposed FedEx-LoRA addresses this by introducing a residual error term added to the frozen weight matrix, ensuring precise updates. FedEx-LoRA is designed to be computationally and communication-efficient, preserving LoRA’s benefits without adding additional training burdens. Empirical results demonstrate improved performance on NLP tasks over SOTA federated fine-tuning methods, highlighting its applicability in distributed training settings where data privacy is a concern.

**Strengths:**

- Originality: The paper introduces a novel solution to the inexact aggregation problem in federated fine-tuning with LoRA by adding a residual error term directly to the pretrained weight matrix. This innovative approach ensures exact updates while preserving the low-rank efficiency of LoRA, addressing a key limitation in existing methods.

- Quality: The authors provide thorough theoretical justification and extensive empirical evaluations across multiple benchmarks. The experiments consistently demonstrate that FedEx-LoRA outperforms SOTA federated fine-tuning methods, validating the effectiveness and robustness of the proposed approach.

- Clarity: The paper is well-written and organized, clearly explaining the challenges of federated fine-tuning and the limitations of existing aggregation methods. The step-by-step presentation of FedEx-LoRA, along with illustrative diagrams and extensive formulas, makes the methodology accessible and easy to understand.

- Significance: By enabling exact aggregation in federated settings without significant computational overhead, the paper has substantial practical implications for privacy-preserving applications in NLP and beyond. The method enhances the feasibility and performance of federated fine-tuning in real-world scenarios where data privacy and efficient communication are crucial.

**Weaknesses:**

- Model limitations: The paper evaluates FedEx-LoRA primarily on RoBERTa and GPT-2, which are smaller foundation models. It remains uncertain how this method performs on larger models, such as Llama and Mistral, where scalability challenges might differ.

- Task scope: The evaluation focuses on standard NLP tasks. Assessing performance on more complex tasks, such as reasoning and inference, would strengthen the paper’s claims on generalizability and robustness.

- Quantization challenges: Introducing residual matrices to the original weights could complicate quantization processes for LLMs, as these residuals require re-quantization at each aggregation round. This could increase computational and memory costs, potentially limiting practical efficiency in federated settings.

- Privacy impacts: Although FedEx-LoRA is presented as beneficial for privacy-preserving federated learning, the paper lacks a detailed analysis of how the approach performs under differential privacy constraints or other mechanisms, which could be helpful for deployment in sensitive domains.

**Questions:**

- What factors contribute to FedEx-LoRA outperforming centralized LoRA on certain benchmarks?

- Can FedEx-LoRA be extended to other LoRA variants, such as MoE-LoRA and QLoRA, and if so, what adaptations might be necessary?

- How does FedEx-LoRA scale and perform with larger foundation models, such as Llama or Gemma?

- How does FedEx-LoRA perform on more complex tasks, such as commonsense QA, mathematic reasoning, or coding?

---

> ### Author Response · Authors · 2024-11-20
> **Reviewer eEFJ Response**
>
> We thank the reviewer for recognizing the strengths of our paper and providing valuable suggestions. We have updated the manuscript to address the concerns raised.
>
>
> **Model Limitations and Task Scope**
>
> How does FedEx-LoRA scale and perform with larger foundation models and complex tasks?
>
> We appreciate these suggestions and have thoroughly addressed these points in our revision. We have expanded our work to evaluate larger models and more complex tasks. Our method **significantly** outperforms state-of-the-art federated LoRA approaches when evaluated on complex tasks with LLMs. This improvement is much more pronounced compared to our previous evaluations with smaller models. The manuscript now includes detailed evaluations of models ranging from 3B to 9B parameters, covering both arithmetic and commonsense reasoning tasks. For arithmetic reasoning, we fine-tuned Mistral-7B and Gemma-2 9B on a subset of the MetaMathQA dataset and evaluated their performance using the GSM8K and MATH benchmarks. For commonsense reasoning, we fine-tuned Llama-3.2 3B on COMMONSENSE170K—a collection of eight distinct commonsense reasoning datasets—and reported performance for each task. The results, presented in Tables 1 and 2 (of the revised manuscript), demonstrate that our method consistently outperforms state-of-the-art federated fine-tuning techniques across all arithmetic benchmarks and commonsense reasoning tasks. For example, FedEx-LoRA achieves an 8.64% improvement over FFA-LoRA and a 2.42% gain over FedIT in average accuracy on commonsense reasoning tasks.
>
>
> Table 1: Results for Llama-3.2 3B on eight commonsense reasoning datasets, comparing various federated LoRA methods at rank r=32.
>
> | Method | BoolQ | PIQA | SIQA | HellaS. | WinoG. | ARC-e | ARC-c | OBQA | Avg. |
> |--------|--------|--------|--------|-----------|---------|---------|---------|--------|--------|
> | Centralized LoRA (r=32) | 73.45 | 89.65 | 82.23 | 94.41 | 87.97 | 93.88 | 82.76 | 86.60 | 86.37 |
> | FedIT (r=32) | 70.73 | 87.59 | 79.17 | 91.06 | 83.42 | 92.71 | 81.31 | 82.68 | 83.57 |
> | FFA-LoRA (r=32) | 65.78 | 84.22 | 72.41 | 82.27 | 72.53 | 90.36 | 76.28 | 75.00 | 77.35 |
> | FedEx-LoRA (r=32) | **73.21** | **89.01** | **81.98** | **94.29** | **87.29** | **93.68** | **82.33** | **86.20** | **85.99** |
>
> Table 2: Arithmetic reasoning performance on GSM8K and MATH for Mistral-7B and Gemma-2 9B, comparing various federated LoRA methods at rank r=32.
>
> | Model        | Method                  | GSM8K     | MATH     |
> |--------------|------------------------|-----------|----------|
> | Mistral-7B   | Centralized LoRA (r=32)| 62.77     | 16.24    |
> |              | FedIT (r=32)           | 56.94     | 14.96    |
> |              | FFA-LoRA (r=32)        | 56.41     | 14.88    |
> |              | FedEx-LoRA (r=32)      | **62.62** | **16.54**|
> |--------------|------------------------|-----------|----------|
> | Gemma-2 9B   | Centralized LoRA (r=32)| 76.34     | 39.32    |
> |              | FedIT (r=32)           | 74.57     | 37.16    |
> |              | FFA-LoRA (r=32)        | 75.04     | 35.18    |
> |              | FedEx-LoRA (r=32)      | **76.19** | **39.00**|
>
>
>
> **Question 1**
>
> What factors contribute to FedEx-LoRA outperforming centralized LoRA on certain benchmarks?
>
> Training on smaller, distributed datasets through federated learning can sometimes prevent the model from overfitting, which may explain why it occasionally performs better than training on the complete centralized dataset.
>
> **Question 2**
>
> Can FedEx-LoRA be extended to other LoRA variants, such as MoE-LoRA and QLoRA, and if so, what adaptations might be necessary?
>
> FedEx-LoRA integrates easily with QLoRA through a simple process. Each client first trains a model using QLoRA and then converts the trained weights to LoRA format using bfloat16 quantization. These converted weights are then sent to the server. The server aggregates the LoRA weights (including the residual matrix) and sends them back to the clients. Each client then loads the aggregated weights into their base model, converts the model back to QLoRA format, and continues training. The quantization and dequantization operations occur entirely on the client side. These operations are negligible compared to the costs of training a model, and they only take place before and after a federated aggregation round, not during the actual training phase.
>
> MoE-LoRA can be used without modifications, requiring only that client models have the same number of LoRA experts. While the effectiveness of the aggregated model remains uncertain when LoRA experts may be trained on different tasks, we emphasize that FedEx-LoRA extends to any LoRA variant compatible with vanilla Federated LoRA, as our contribution lies in the aggregation mechanism.
>
> **Privacy Impacts**
>
> We thank the reviewer for pointing this out. This is indeed an interesting area of study, but we have focused our current work elsewhere and plan to explore this thoroughly in future studies.

---

> > ### Author Response · Authors · 2024-11-24
> > **Reminder**
> >
> > Thank you for your feedback so far. We have provided a detailed rebuttal and an updated draft addressing your comments with additional results. Please feel free to share any further questions or thoughts before the discussion period ends.

---

> > ### Comment · Reviewer_eEFJ · 2024-11-26
> > **Reply to Author Response**
> >
> > Thank you for addressing the concerns and providing additional evaluations, which add valuable depth to the paper. While I appreciate the improvements, I believe it is appropriate to maintain the current positive assessment.

---

> > > ### Author Response · Authors · 2024-11-28
> > > **Response**
> > >
> > > Thank you for your positive review. We would like to emphasize an additional contribution: our novel communication protocol for hyperclient settings. We have validated this through new experimental results:
> > >
> > > We fine-tuned Mistral-7B with 100 clients on a subset of MetaMathQA and evaluated on GSM8K and MATH benchmarks - scenarios where the original method would incur prohibitive communication costs. Since communication cost scales linearly with the truncated SVD rank of the residual matrix and equals FedIT's cost at the same rank, our results directly demonstrate the efficiency-performance trade-off:
> > >
> > > | Method | Rank | GSM8K | MATH |
> > > |--------|------|--------|-------|
> > > | FedIT | 32 | 54.89 | 13.64 |
> > > | FedEx | 32 | **54.96** | **14.02** |
> > > | FedEx | 16 | 54.36 | 13.84 |
> > >
> > > These results show that our method not only outperforms FedIT at equal communication bandwidth but maintains similar performance even when using half the bandwidth. This demonstrates that our communication protocol effectively addresses the overhead concerns while improving upon baseline efficiency.

---

### Official Review · Reviewer_Kh7U · 2024-11-02

**Soundness:** 3
**Presentation:** 4
**Contribution:** 2
**Rating:** 5
**Confidence:** 4

**Summary:**

This paper explores the task of parameter-efficient fine-tuning under Federated Learning setting, particularly focused on the low rank adaptation approach. The work is built on the insights from the earlier FedIT model which used traditional federated aggregation methods to average low-rank matrices A and B individually. However, this approach was found to be inexact aggregation as the average of the products does not equate to the product of the averages.
To address this challenge, the authors propose an innovative method that incorporates a residual term into the pretrained frozen weight matrix. This addition aims to bridge the discrepancy between the average of the products and the product of the averages, enhancing the aggregation performance. The effectiveness of this approach is demonstrated through extensive analysis of NLU and NLG benchmark datasets using RoBERTa and GPT-2 models. The results show notable improvements in performance metrics while maintaining comparably low communication costs compared to the most benchmarked models.

**Strengths:**

The paper introduces a straightforward yet effective method that ensures exact aggregation in LoRA-based fine tuning for Federated Learning. While it has a slightly higher communication cost compared to FedIT and FFA-LoRA, the increase is minimal given the significant improvement in overall performance.
Moreover, the method's effectiveness is validated through comprehensive experiments and analysis, affirming its claim of exact aggregation and subsequently better overall performance.
Additionally, the paper has good reproducibility of the results because of its detailed presentation of the experimental settings.

**Weaknesses:**

The main issue is that the contribution may seem too incremental for ICLR conference as the method primarily focuses on a straightforward adjustment in the aggregation phase—specifically, adding back the discrepancy between the average of the products and the product of the averages. This method, while empirically effective, does not introduce a significant innovation or a novel approach that would typically be expected for ICLR.
Additionally, there are practical scalability concerns regarding the residue error term matrix, which shares dimensions with W’ and W0. The paper notes that the rank of this matrix is limited to k*r where k is the number of clients. However, in typical federated learning environments, which often involve a large number of clients (potentially hundreds), this could lead to a high-rank matrix and, consequently, significant communication costs. The experimental setup in the paper, which involved only three clients, does not adequately demonstrate the method's communication cost in larger, more realistic scenarios common in federated learning.

**Questions:**

1.	It would be good to see the performance comparison for different settings regarding clients e.g., the number of clients,  the portion of randomly selected clients etc. – these are standard evaluations in FL literature
2.	How the residue error term matrix affects the update of A and B? It would be good to see an analysis experiment regarding this.

---

> ### Author Response · Authors · 2024-11-20
> **Reviewer Kh7U Response**
>
> We thank the reviewer for the constructive feedback. We have carefully addressed each concern and question raised in the review.
>
> **Weaknesses**
>
> **Novelty**
>
> While the method is straightforward, we believe this simplicity is a strength rather than a limitation. Like widely-adopted methods such as AdamW (Loshchilov, et al.,2019),  FedEx-LoRA's value lies in its practical utility and ease of implementation while providing significant empirical improvements.
>
> **Scalability**
>
> We acknowledge the scaling concerns with larger client populations. We propose using truncated SVD, which maintains the same communication cost as FedIT while providing optimal approximation:
>
> Best Inexact Approximation
>
> For exact aggregation, the communication cost scales linearly with the number of clients, becoming prohibitive in hyperclient settings. To address this, we propose relaxing the exact aggregation condition through truncated SVD of the residual matrix. This reconstruction yields a low-rank approximation which, by the Eckart-Young theorem, is provably optimal for the high-rank update matrix. Specifically, for a target rank r', the best low-rank approximation ΔW_rec' is computed as:
>
> U, S, V.T ← SVD(ΔW_res),
>  ΔW_rec' ← U[1 : r'] S[1 : r', 1 : r'] V.T[1 : r']
>
> While this method introduces approximation error, it provides the theoretically optimal approximation to exact aggregation. A key advantage is that the server can control communication costs, a capability absent in previous methods - FedIT and FFA-LoRA.
> The method has also been updated in the manuscript (Section 4, lines 249-261).
>
>
>
>
>
> To test this efficient communication protocol, we evaluate our method in scenarios with a large number of clients where the original method would incur prohibitive communication costs.
>
> We fine-tune Mistral-7B with 100 clients on a subset of MetaMathQA and evaluate on GSM8K and MATH benchmarks. Since communication cost scales linearly with the truncated SVD rank of the residual matrix and equals FedIT's cost at the same rank, our results are directly comparable in terms of communication overhead. The table below shows these results, where 'Rank' indicates the rank of the communicated matrix. Our method outperforms FedIT at equal communication bandwidth and maintains similar performance even when using half the bandwidth.
>
>
> | Method | Rank | GSM8K | MATH |
> |--------|------|--------|-------|
> | FedIT | 32 | 54.89 | 13.64 |
> | FedEx | 32 | **54.96** | **14.02** |
> | FedEx | 16 | 54.36 | 13.84 |
>
>
> **Questions**
>
> 1. To assess performance across different settings, we expand our evaluation with additional experiments using 10 clients, randomly selecting 20% per round, and applying truncated SVD with rank 4 for aggregation. The results show promising scalability: FedEx-LoRA improves RoBERTa-base performance on CoLA from 54.17 to 55.26 and on MRPC from 85.21 to 86.27, outperforming FedIT. These findings suggest that our method maintains its advantages even with more clients, while keeping communication costs comparable to FedIT. Additionally, it achieves the theoretically optimal approximation, as evidenced by a reduction in the scaled Frobenius norm of deviation from 0.26 to 0.075 (this metric is clearly defined in Section 6 of the paper).
> Our experimental setup follows established protocols from recent federated LoRA works (Sun et al., 2024; Cho et al., 2024) to ensure fair comparisons. However, we agree that evaluating our method in diverse settings would help demonstrate its robustness across various scenarios.
>
>
>
>
> 2. The effect of the residual term can be indirectly measured or analyzed. Since the residual term added to the product of the aggregated matrices A and B is constant, modifying the residual term effectively changes the assignment of A and B after aggregation. In this context, we believe that the discussion in Section 6 on “Assignment Strategies for A and B” is exactly equivalent to analyzing the impact of varying the residual term.
>
> [1] Loshchilov, I., & Hutter, F. (2019). Decoupled Weight Decay Regularization. arXiv preprint arXiv:1711.05101.
>
>
> [2] Sun et al. "Improving LoRA in Privacy-preserving Federated Learning",
> arXiv:2403.12313, 2024
>
>
> [3] Cho et al. "Heterogeneous LoRA for Federated Fine-tuning of On-Device Foundation Models", arXiv:2401.06432, 2024

---

> > ### Author Response · Authors · 2024-11-24
> > **Reminder**
> >
> > We thank the reviewer for their comments. We’ve carefully revised the paper based on your comments and included additional findings in the updated draft shared earlier. Please let us know if there are any outstanding questions or areas that need further clarification as we approach the end of the discussion period.

---

### Official Review · Reviewer_kdXw · 2024-11-02

**Soundness:** 2
**Presentation:** 3
**Contribution:** 2
**Rating:** 5
**Confidence:** 5

**Summary:**

This work provide a method, called FedEx-LoRA, to address the inexact aggregation problem when applying the LoRA in federated learning environment for large language model fine-tuning. FedEx-LoRA calculates a residual error matrix to the frozen pertained matrix on the server and then send it to each client to correct the error bring by the LoRA matrix aggregation on the server. This idea is straightforward. The authors also doing some experiments based on the LLM benchmarks to evaluate the model accuracy.

**Strengths:**

This work provides a method, called FedEx-LoRA, to address the inexact aggregation problem when applying the LoRA in federated learning environment for large language model fine-tuning.

**Weaknesses:**

There are some concerns for the proposed methods:

1) for the federated learning, the network bandwidth between the server and clients is often very limited. This work brings lots of extra communication overhead for each training round. The extra communication data between server and clients equals to the entire model size which brings strong communication overhead, especially for the federated learning environments.

2) the proposed methods requires to calculate n matrix multiplication, where n is the number of the layers for the LLM models. For the federated learning, the server also run on a CPU-based computing node. The n matrix multiplication also brings strong computational overhead for the server. If you want to use GPU to calculate such matrix multiplication, you need to carefully schedule the calculation process since the bandwidth between main memory and GPU memory (PCIe) also limited.

Overall, the final system performance, including the communication and computational efficiency for the proposed method is unacceptable for the federated learning environments.

**Questions:**

Please see the comments from the above weakness section. Two questions need to answer:

1) how to address the strong communication overhead problem between the server and clients?

2) how to address the strong computation overhead problem on the server?

These two issues will largely cause system performance degradation.

---

> ### Author Response · Authors · 2024-11-20
> **Reviewer kdXw Response**
>
> We thank the reviewer for the constructive suggestions. We have addressed each point and concern raised in the review.
>
> **Communication Cost**
>
> The communication overhead is carefully managed through matrix decomposition. Specifically, since ∆Wres has rank ≤ (kr), we can decompose it using Singular Value Decomposition (SVD) into U, S, V. Now the singular values will be 0 post row and column > (kr). We can simply take the first (kr) rows of U and similar columns for V. This decomposition means we only need to transmit matrices of size m×(kr) and (kr)×n, requiring (kr)(m+n) parameters. In contrast, transmitting the full weight update would require m×n parameters.
>
> For concrete quantification, consider our RoBERTa-base experiments where m=n=768, k=3 clients, and r=4: FedEx-LoRA requires transmitting only 2×768×(3×4) = 18,432 parameters per layer, compared to 768×768 = 589,824 parameters for full model updates. This represents a 32x reduction in communication cost per layer (the total cost is proportional to the cost per layer). This analysis aligns with our empirical measurements in Table 4 (original manuscript) or Table 6 (revised manuscript), where FedEx-LoRA shows only marginal overhead compared to standard federated LoRA approaches while being significantly cheaper than federated full fine-tuning.
>
>
> **Computation Cost**
>
> The matrix multiplications required for aggregation are minimal compared to the computational demands of model training on clients. For perspective, a single forward pass by any client requires orders of magnitude more computation than the server's aggregation operations. The training computations dwarf the simple matrix operations needed for aggregation.
> Recent work in federated learning (Kairouz, et al.,2021) advocates for leveraging server computation to improve system efficiency. We believe that our minimal computational overhead should not be considered a weakness as it aligns well with modern FL approaches.
>
>
> References:
>
> [1] Li, Z., Yan, C., Zhang, X., Gharibi, G., Yin, Z., Jiang, X., & Malin, B. A. (2023). Split Learning for Distributed Collaborative Training of Deep Learning Models in Health Informatics. arXiv preprint arXiv:2308.11027.
>
> [2] P. Kairouz, H. B. McMahan, B. Avent, et al., "Advances and open problems in federated learning," arXiv preprint arXiv:1912.04977, 2021.

---

> > ### Comment · Reviewer_kdXw · 2024-11-26
> >
> > Thanks for the rebuttal from the authors. Thanks for the clarification regarding the contribution. However, there are still some concerns for me. Such as, using the SVD to decompose the matrix also brings strong computational overhead for the server since you need to use the SVD to decompose each layer of the model, especially for the model with a large number of layers. SVD is a computation-intensive method. In addition, using SVD to decompose the matrix also involves some errors in the training, potentially affects training performance, and decreases the benefit that is achieved by the proposed FedEx-LoRA approach.
> >
> > Overall, I think the authors' proposed solution is not a solid way to address the issues I noticed, so I will maintain my score.

---

> > > ### Author Response · Authors · 2024-11-28
> > > **Response**
> > >
> > > We thank the reviewer for their response to the updated draft and rebuttal. Let us address each concern systematically:
> > >
> > > Regarding SVD approximation errors: While the reviewer's point about SVD introducing errors is valid, we note that truncated SVD provides the theoretically optimal low-rank approximation to exact updates. To validate that this approximation maintains performance, we conducted extensive experiments with our efficient communication protocol.
> > >
> > > We fine-tuned Mistral-7B with 100 clients on a subset of MetaMathQA and evaluated on GSM8K and MATH benchmarks - scenarios where the original method would incur prohibitive communication costs. Since communication cost scales linearly with the truncated SVD rank of the residual matrix and equals FedIT's cost at the same rank, our results directly demonstrate the efficiency-performance trade-off:
> > >
> > > | Method | Rank | GSM8K | MATH |
> > > |--------|------|--------|-------|
> > > | FedIT | 32 | 54.89 | 13.64 |
> > > | FedEx | 32 | **54.96** | **14.02** |
> > > | FedEx | 16 | 54.36 | 13.84 |
> > >
> > > These results show that our method not only outperforms FedIT at equal communication bandwidth but maintains similar performance even when using half the bandwidth. This demonstrates that our communication protocol effectively addresses the overhead concerns while improving upon baseline efficiency.
> > >
> > > Regarding computational overhead of SVD: While we acknowledge the increased computational requirement, several key points demonstrate why this is not a significant limitation:
> > >
> > > 1. The SVD computation is performed once per layer post-aggregation and crucially does not scale with the number of clients. This fixed overhead is more efficient than computing full SVD since only top vectors are needed.
> > >
> > > 2. Our approach is computationally more efficient than existing methods in federated LoRA literature:
> > >    - In [1], the server performs ADAM optimization steps, which are orders of magnitude more computationally expensive than truncated SVD per layer
> > >    - In [2], the server calculates full SVD decomposition for each client, resulting in client-linear scaling (Their Frobenius norm approach for efficient SVD can be directly adapted to our method)
> > >    - In [3], the server aggregates full finetuning matrices before SVD for adapter initialization, incurring comparable computational costs
> > >
> > > These established works in literature demonstrate that server-side computation overhead is an acceptable trade-off when it enables improved communication efficiency and performance in federated LoRA settings. Our method maintains this principle while providing better tunable communication-performance trade-offs.
> > >
> > > [1] Kuo et al. "Federated LoRA with Sparse Communication", arXiv:2406.05233, 2024
> > >
> > > [2] Cho et al. "Heterogeneous LoRA for Federated Fine-tuning of On-Device Foundation Models", arXiv:2401.06432, 2024
> > >
> > > [3] S. Babakniya, A. Elkordy, Y. Ezzeldin, Q. Liu, K.-B. Song, M. EL-Khamy, and S. Avestimehr. SLoRA: Federated parameter efficient fine-tuning of language models. In International Workshop on Federated Learning in the Age of Foundation Models in Conjunction with NeurIPS 2023, 2023. URL https://openreview.net/forum?id=06quMTmtRV.

---

> ### Author Response · Authors · 2024-11-24
> **Reminder**
>
> Thank you for your comments and questions. We hope the rebuttal adequately addresses your feedback. Additionally, we’ve included new results in the revised version and would appreciate your thoughts or further discussion on this before the discussion period concludes.

---

### Author Response · Authors · 2024-11-20
**Updates and Major Concerns**

We appreciate the reviewers' constructive feedback and have made revisions to the manuscript to address their concerns. Below, we highlight the key changes and improvements incorporated in response to the reviewers' comments.

**Arithmetic Reasoning and Commonsense Reasoning Experiments**


We have expanded our work to evaluate larger models and more complex tasks. Our method **significantly** outperforms state-of-the-art federated LoRA approaches when evaluated on complex tasks with LLMs. This improvement is much more pronounced compared to our previous evaluations with smaller models. The manuscript now includes detailed evaluations of models ranging from 3B to 9B parameters, covering both arithmetic and commonsense reasoning tasks. For arithmetic reasoning, we fine-tuned Mistral-7B and Gemma-2 9B on a subset of the MetaMathQA dataset and evaluated their performance using the GSM8K and MATH benchmarks. For commonsense reasoning, we fine-tuned Llama-3.2 3B on COMMONSENSE170K—a collection of eight distinct commonsense reasoning datasets—and reported performance for each task. The results, presented in Tables 1 and 2 (of the revised manuscript), demonstrate that our method consistently outperforms state-of-the-art federated fine-tuning techniques across all arithmetic benchmarks and commonsense reasoning tasks. For example, FedEx-LoRA achieves an 8.64% improvement over FFA-LoRA and a 2.42% gain over FedIT in average accuracy on commonsense reasoning tasks.



Table 1: Results for Llama-3.2 3B on eight commonsense reasoning datasets, comparing various federated LoRA methods at rank r=32. Centralized LoRA sets the benchmark skyline for its federated versions. Best results among federated methods are in bold.

| Method | BoolQ | PIQA | SIQA | HellaS. | WinoG. | ARC-e | ARC-c | OBQA | Avg. |
|--------|--------|--------|--------|-----------|---------|---------|---------|--------|--------|
| Centralized LoRA (r=32) | 73.45 | 89.65 | 82.23 | 94.41 | 87.97 | 93.88 | 82.76 | 86.60 | 86.37 |
| FedIT (r=32) | 70.73 | 87.59 | 79.17 | 91.06 | 83.42 | 92.71 | 81.31 | 82.68 | 83.57 |
| FFA-LoRA (r=32) | 65.78 | 84.22 | 72.41 | 82.27 | 72.53 | 90.36 | 76.28 | 75.00 | 77.35 |
| FedEx-LoRA (r=32) | **73.21** | **89.01** | **81.98** | **94.29** | **87.29** | **93.68** | **82.33** | **86.20** | **85.99** |

Table 2: Arithmetic reasoning performance on GSM8K and MATH for Mistral-7B and Gemma-2 9B, comparing various federated LoRA methods at rank r=32. Centralized LoRA sets the benchmark skyline for its federated versions. Best results among federated methods are in bold.

| Model        | Method                  | GSM8K     | MATH     |
|--------------|------------------------|-----------|----------|
| Mistral-7B   | Centralized LoRA (r=32)| 62.77     | 16.24    |
|              | FedIT (r=32)           | 56.94     | 14.96    |
|              | FFA-LoRA (r=32)        | 56.41     | 14.88    |
|              | FedEx-LoRA (r=32)      | **62.62** | **16.54**|
|--------------|------------------------|-----------|----------|
| Gemma-2 9B   | Centralized LoRA (r=32)| 76.34     | 39.32    |
|              | FedIT (r=32)           | 74.57     | 37.16    |
|              | FFA-LoRA (r=32)        | 75.04     | 35.18    |
|              | FedEx-LoRA (r=32)      | **76.19** | **39.00**|


**Efficient Communication for Hyperclient Settings**

For exact aggregation, the communication cost scales linearly with the number of clients, becoming prohibitive in hyperclient settings. To address this, we propose relaxing the exact aggregation condition through truncated SVD of the residual matrix. This reconstruction yields a low-rank approximation which, by the Eckart-Young theorem, is provably optimal for the high-rank update matrix. While this method introduces approximation error, it provides the theoretically optimal approximation to exact aggregation. A key advantage is that the server can control communication costs, a capability absent in previous methods. We provide a discussion of this approach and its benefits in Section 4 of the revised manuscript.

---

### Note · Authors · 2024-12-15

I have read and agree with the venue's withdrawal policy on behalf of myself and my co-authors.